# Pyramidal Flow Matching for Efficient Video Generative Modeling

**Yang Jin[1], Zhicheng Sun[1], Ningyuan Li[3], Kun Xu, Kun Xu[2], Hao Jiang[1], Nan Zhuang[2],**
**Quzhe Huang, Yang Song, Yadong Mu[1]\*, Zhouchen Lin[4,5,6]\***
[1]Peking University, [2]Kuaishou Technology, [3]Beijing University of Posts and Telecommunications,
[4]State Key Lab of General AI, School of Intelligence Science and Technology, Peking University,
[5]Institute for Artificial Intelligence, Peking University,
[6]Pazhou Laboratory (Huangpu), Guangzhou, Guangdong, China

## Abstract

Video generation requires modeling a vast spatiotemporal space, which demands significant computational resources and data usage. To reduce the complexity, the prevailing approaches employ a cascaded architecture to avoid direct training with full resolution latent. Despite reducing computational demands, the separate optimization of each sub-stage hinders knowledge sharing and sacrifices flexibility. This work introduces a unified pyramidal flow matching algorithm. It reinterprets the original denoising trajectory as a series of pyramid stages, where only the final stage operates at the full resolution, thereby enabling more efficient video generative modeling. Through our sophisticated design, the flows of different pyramid stages can be interlinked to maintain continuity. Moreover, we craft autoregressive video generation with a temporal pyramid to compress the full-resolution history. The entire framework can be optimized in an end-to-end manner and with a single unified Diffusion Transformer (DiT). Extensive experiments demonstrate that our method supports generating high-quality 5-second (up to 10-second) videos at 768p resolution and 24 FPS within 20.7k A100 GPU training hours. All code and models are open-sourced at `https://pyramid-flow.github.io`.

## 1 Introduction

Video is a media form that records the evolvement of the physical world. Teaching the AI system to generate various video content plays a vital role in simulating the real-world dynamics (Hu et al., 2023; Brooks et al., 2024) and interacting with humans (Bruce et al., 2024; Valevski et al., 2024). Nowadays, the cutting-edge diffusion models (Ho et al., 2022c; Blattmann et al., 2023a; OpenAI, 2024) and autoregressive models (Yan et al., 2021; Hong et al., 2023; Kondratyuk et al., 2024) have made remarkable breakthroughs in generating realistic and long-duration video through scaling of data and computation. However, the necessity of modeling a significantly large spatiotemporal space makes the training of such video generative models computationally and data intensive.

To ease the computational burden of generating high-dimensional video data, a crucial component is to compress the original video pixels into a lower-dimensional latent space using a VAE (Kingma & Welling, 2014; Esser et al., 2021; Rombach et al., 2022). However, the regular compression rate (typically $8\times$) still results in excessive tokens, especially for high-resolution samples. In light of this, prevalent approaches utilize a cascaded architecture (Ho et al., 2022b; Pernias et al., 2024; Teng et al., 2024) to break down the high-resolution generation process into multiple stages, where samples are first created in a highly compressed latent space and then successively upsampled using additional super-resolution models. Although the cascaded pipeline avoids directly learning at high resolution and reduces the computational demands, the requirement for employing distinct models at different resolutions separately sacrifices flexibility and scalability. Besides, the separate optimization of multiple sub-models also hinders the sharing of their acquired knowledge.

This work presents an efficient video generative modeling framework that transcends the limitations of the previous cascaded approaches. Our motivation stems from the observation in Fig. 1a that the initial timesteps in diffusion models are quite noisy and uninformative. This suggests that operating at full resolution throughout the entire generation trajectory may not be necessary. To this end, we reinterpret the original generation trajectory as a series of pyramid stages that operate on compressed

---

\*Yadong Mu and Zhouchen Lin are corresponding authors.

Frame index:    $i = 0$     $i = 1$     $i = T$        $i = 0$     $i = 1$     $i = T$

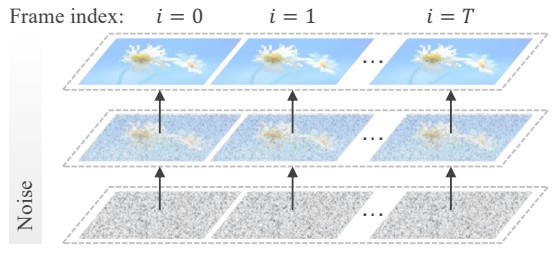 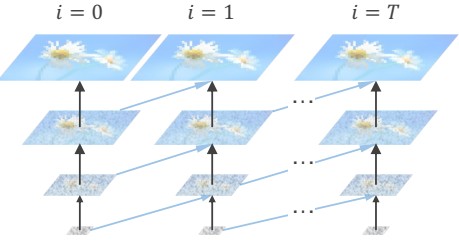

Noise

     (a) Video diffusion model like Sora (OpenAI, 2024)         (b) Our proposed pyramidal flow matching

Figure 1: A motivating example for pyramidal flow matching: (a) Existing diffusion models operate at full resolution, spending a lot of computation on very noisy latents. (b) Our method harnesses the flexibility of flow matching to interpolate between latents of different resolutions. This allows for simultaneous generation and decompression of visual content with better computational efficiency. Note that the black arrows are denoising trajectories, and the blue ones are their temporal conditions.

representations of different scales. Notably, the efficacy of image pyramids (Adelson et al., 1984) has been widely validated for discriminative neural networks (Lin et al., 2017; Wang et al., 2020) and more recently for diffusion models (Ho et al., 2022b; Pernias et al., 2024; Teng et al., 2024)and multimodal LLMs (Yu et al., 2023; Tian et al., 2024). Here, we investigate two types of pyramids: the spatial pyramid within a frame and the temporal one between consecutive frames (as illustrated in Fig. 1b). In such a pyramidal generation trajectory, only the final stage operates at full resolution, drastically reducing redundant computations in earlier timesteps. The main advantages are twofold: (1) The generation trajectories of different pyramid stages are interlinked, with the subsequent stage continuing to generate from the previous ones. This eliminates the need for each stage to regenerate from pure noise in some cascade models. (2) Instead of relying on separate models for each image pyramid, we integrate them into a single unified model for end-to-end optimization, which admits drastically-expedited training with more elegant implementation as validated by experiments.

Based on the aforementioned pyramidal representations, we introduce a novel pyramidal flow matching algorithm that builds upon recent prevalent flow matching framework (Lipman et al., 2023; Liu et al., 2023; Albergo & Vanden-Eijnden, 2023). Specifically, we devise a piecewise flow for each pyramid resolution, which together form a generative process from noise to data. The flow within each pyramid stage takes a similar formulation, interpolating between a pixelated (compressed) and noisier latent and a pixelate-free (decompressed) and cleaner latent. Through our design, they can be jointly optimized by the unified flow matching objective in a single Diffusion Transformer (DiT) (Peebles & Xie, 2023), allowing simultaneous generation and decompression of visual content without multiple separate models. During inference, the output of each stage is renoised by a corrective Gaussian noise that maintains the continuity of the probability path between stages. Furthermore, we formulate the video generation in an autoregressive manner, iteratively predicting the next video latent conditioned on the generated history. Given the high redundancy in the full-resolution history, we curate a temporal pyramid sequence using progressively compressed, lower-resolution history as conditions, further reducing the number of tokens and improving training efficiency.

The collaboration of the spatial and temporal pyramids results in remarkable training efficiency for video generation. Compared to the commonly used full-sequence diffusion, our method significantly reduces the number of video tokens during training (e.g., $\leq 15{,}360$ tokens versus $119{,}040$ tokens for a 10-second, 241-frame video), thereby reducing both computational resources required and training time. By training only on open-source datasets, our model generate high-quality 10-second videos at 768p resolution and 24 fps. The core contributions of this paper are summarized as follows:

- We present pyramidal flow matching, a novel video generative modeling algorithm that incorporates both spatial and temporal pyramid representations. Utilizing this framework can significantly improve training efficiency while maintaining good video generation quality.

- The proposed unified flow matching objective facilitates joint training of pyramid stages in a single Diffusion Transformer (DiT), avoiding the separate optimization of multiple models. The support for end-to-end training further enhances its simplicity and scalability.

- We evaluate its effectiveness on VBench (Huang et al., 2024) and EvalCrafter (Liu et al., 2024), with highly competitive performance among video generative models trained on public datasets.

## 2   RELATED WORK

**Video Generative Models** have seen rapid progress with autoregressive models (Yan et al., 2021; Hong et al., 2023; Kondratyuk et al., 2024; Jin et al., 2024) and diffusion models (Ho et al., 2022c; Blattmann et al., 2023b;a). A notable breakthrough is the high-fidelity video diffusion models (OpenAI, 2024; Kuaishou, 2024; Luma, 2024; Runway, 2024) by scaling up DiT pre-training (Peebles & Xie, 2023), but they induce significant training costs for long videos. An alternative line of research integrates diffusion models with autoregressive modeling (Chen et al., 2024a; Valevski et al., 2024) to natively support long video generation, but is still limited in context length and training efficiency. Our work advances both approaches in terms of efficiency from a compression perspective, featuring a spatially compressed pyramidal flow and a temporally compressed pyramidal history.

**Image Pyramids** (Adelson et al., 1984) have been studied extensively in visual representation learning (Lowe, 2004; Dalal & Triggs, 2005; Lin et al., 2017; Wang et al., 2020). For generative models, the idea is explored by cascaded diffusion models that first generate at low resolution and then perform super-resolution (Ho et al., 2022b; Saharia et al., 2022; Zhang et al., 2023b; Gu et al., 2023; Pernias et al., 2024; Teng et al., 2024), and later extended to video (Ho et al., 2022a; Singer et al., 2023). However, they require training several separate models, which prevents knowledge sharing. Possible unified modeling solutions for pyramids include hierarchical architectures (Rombach et al., 2022; Crowson et al., 2024; Hatamizadeh et al., 2024) or via next-token prediction (Yu et al., 2023; Tian et al., 2024), but involve architectural changes. Instead, we propose a simple flow matching objective that allows joint training of pyramids, thus facilitating efficient video generative modeling.

## 3   METHOD

This work proposes an efficient video generative modeling scheme named pyramidal flow matching. In the following text, we first extend the flow matching algorithm (Section 3.1) to an efficient spatial pyramid representation (Section 3.2). Then, a temporal pyramid design is proposed in Section 3.3 to further improve training efficiency. Lastly, practical implementations are discussed in Section 3.4.

### 3.1   PRELIMINARIES ON FLOW MATCHING

Similar to diffusion models (Sohl-Dickstein et al., 2015; Song & Ermon, 2019; Ho et al., 2020), flow generative models (Papamakarios et al., 2021; Song et al., 2021; Xu et al., 2022; Lipman et al., 2023; Liu et al., 2023; Albergo & Vanden-Eijnden, 2023) aim to learn a velocity field $v_t$ that maps random noise $x_0 \sim \mathcal{N}(0, I)$ to data samples $x_1 \sim q$ via an ordinary differential equation (ODE):

$$\frac{dx_t}{dt} = v_t(x_t). \tag{1}$$

Recently, Lipman et al. (2023); Liu et al. (2023); Albergo & Vanden-Eijnden (2023) proposed the flow matching framework, which provides a simple simulation-free training objective for flow generative models by directly regressing the velocity $v_t$ on a conditional vector filed $u_t(\cdot|x_1)$:

$$\mathbb{E}_{t,q(x_1),p_t(x_t|x_1)} \big\| v_t(x_t) - u_t(x_t|x_1) \big\|^2, \tag{2}$$

where $u_t(\cdot|x_1)$ uniquely determines a conditional probability path $p_t(\cdot|x_1)$ toward data sample $x_1$. An effective choice of the conditional probability path is linear interpolation of data and noise:

$$x_t = tx_1 + (1-t)x_0, \tag{3}$$

$$x_t \sim \mathcal{N}(tx_1, (1-t)^2 I), \tag{4}$$

and $u(x_t|x_1) = x_1 - x_0$. Notably, flow matching can be flexibly extended to interpolate between distributions other than standard Gaussians. This enables us to devise a new flow matching algorithm that specializes in reducing the computational cost of video generative modeling.

### 3.2   PYRAMIDAL FLOW MATCHING

The main challenge in video generative modeling is the spatio-temporal complexity, and we address its spatial complexity first. According to previous key observation in Fig. 1, the initial generation

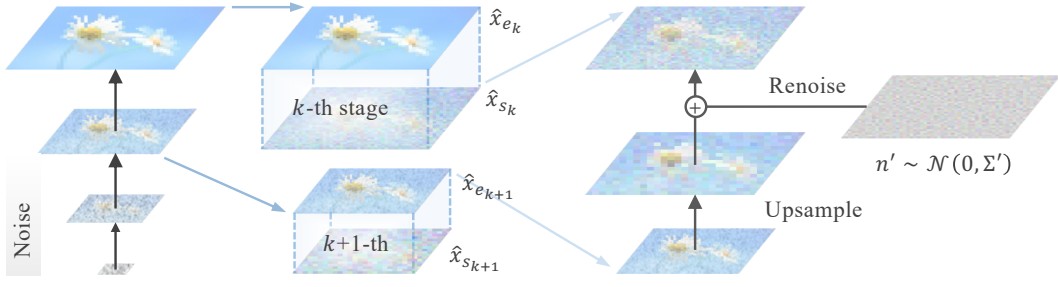

(a) Unified modeling of pyramidal flow      (b) Handling jump points via renoising

Figure 2: Illustration of spatial pyramid. (a) The pyramidal flow is divided into multiple stages, each from a pixelated and noisy starting point to a pixelate-free and cleaner result. (b) During inference, we add a corrective noise at jump points across stages to ensure continuity of the proabability path.

steps are usually very noisy and less informative, and thus may not need to operate at full resolution latent. This motivates us to study a spatially compressed pyramidal flow, illustrated in Fig. 2.

To alleviate redundant computation in early steps, we interpolate flow between data and compressed low-resolution noise. Let $\oplus$ denote the interpolation between latents of different resolutions, and let there be $K$ resolutions, each halving the previous one, then our flow may be expressed as:

$$\hat{\boldsymbol{x}}_t = t\boldsymbol{x}_1 \oplus (1 - t)\,Down(\boldsymbol{x}_0, 2^K), \tag{5}$$

where $Down(\cdot, \cdot)$ is a downsampling function. Since the interpolation concerns varying-dimensional $\boldsymbol{x}_t$, we decompose it as a piecewise flow (Yan et al., 2024) that divides $[0, 1]$ into $K$ time windows, where each window interpolates between successive resolutions. For the $k$-th time window $[s_k, e_k]$, let $t' = (t - s_k)/(e_k - s_k)$ denote the rescaled timestep, then the flow within it follows:

$$\hat{\boldsymbol{x}}_t = t'\,Down(\boldsymbol{x}_{e_k}, 2^k) + (1 - t')\,Up(Down(\boldsymbol{x}_{s_k}, 2^{k+1})), \tag{6}$$

where $Up(\cdot)$ is an upsampling function. This way, only the last stage is performed at full resolution, while most stages are performed at lower resolutions using less computation. Under a uniform stage partitioning, the idea of spatial pyramid reduces the computational cost to a factor of nearly $1/K$. Below, we describe the instantiation of pyramidal flow from training and inference, respectively.

### 3.2.1 UNIFIED TRAINING

In the construction of pyramidal flow, our main concern is unified modeling of different stages, as previous works (Ho et al., 2022b; Pernias et al., 2024; Teng et al., 2024) all require training multiple models for separate generation and super-resolution, which hinders knowledge sharing.

To unify the objectives of generation and decompression/super-resolution, we curate the probability path by interpolating between different noise levels and resolutions. It starts with a more noisy and pixelated latent upsampled from a lower resolution, and yields cleaner and fine-grained results at a higher resolution, as illustrated in Fig. 2a. Formally, the conditional probability path is defined by:

$$\text{End:} \quad \hat{\boldsymbol{x}}_{e_k}|\boldsymbol{x}_1 \sim \mathcal{N}(e_k\,Down(\boldsymbol{x}_1, 2^k), (1 - e_k)^2\boldsymbol{I}), \tag{7}$$

$$\text{Start:} \quad \hat{\boldsymbol{x}}_{s_k}|\boldsymbol{x}_1 \sim \mathcal{N}(s_k\,Up(Down(\boldsymbol{x}_1, 2^{k+1})), (1 - s_k)^2\boldsymbol{I}), \tag{8}$$

where $s_k < e_k$, and the upsampling and downsampling functions for the clean $\boldsymbol{x}_1$ are well defined, *e.g.*, by nearest or bilinear resampling. In addition, to enhance the straightness of the flow trajectory, we couple the sampling of its endpoints by enforcing the noise to be in the same direction. Namely, we first sample a noise $\boldsymbol{n} \sim \mathcal{N}(\boldsymbol{0}, \boldsymbol{I})$ and then jointly compute the endpoints $(\hat{\boldsymbol{x}}_{e_k}, \hat{\boldsymbol{x}}_{s_k})$ as:

$$\text{End:} \quad \hat{\boldsymbol{x}}_{e_k} = e_k\,Down(\boldsymbol{x}_1, 2^k) + (1 - e_k)\boldsymbol{n}, \tag{9}$$

$$\text{Start:} \quad \hat{\boldsymbol{x}}_{s_k} = s_k\,Up(Down(\boldsymbol{x}_1, 2^{k+1})) + (1 - s_k)\boldsymbol{n}. \tag{10}$$

Thereafter, we can regress the flow model $\boldsymbol{v}_t$ on the conditional vector field $\boldsymbol{u}_t(\hat{\boldsymbol{x}}_t|\boldsymbol{x}_1) = \hat{\boldsymbol{x}}_{e_k} - \hat{\boldsymbol{x}}_{s_k}$ with the following flow matching objective to unify generation and decompression:

$$\mathbb{E}_{k,t,(\hat{\boldsymbol{x}}_{e_k}, \hat{\boldsymbol{x}}_{s_k})} \left\| \boldsymbol{v}_t(\hat{\boldsymbol{x}}_t) - (\hat{\boldsymbol{x}}_{e_k} - \hat{\boldsymbol{x}}_{s_k}) \right\|^2. \tag{11}$$

---

**Algorithm 1** Sampling with Pyramidal Flow Matching

---

**Require:** flow model $\boldsymbol{v}$, number of stages $K$, time windows $[s_k, e_k]$.
    Initialize a starting point $\hat{\boldsymbol{x}}_0 \sim \mathcal{N}(\mathbf{0}, \boldsymbol{I})$.
    **for** $k \leftarrow K - 1$ to $0$ **do**
        Compute endpoint $\hat{\boldsymbol{x}}_{e_k}$ from starting point $\hat{\boldsymbol{x}}_{s_k}$ based on the flow model $\boldsymbol{v}$.
        Compute next starting point by upsampling $\hat{\boldsymbol{x}}_{e_k}$ with renoising.        ▷ Eq. (15)
**Ensure:** generated sample $\hat{\boldsymbol{x}}_1$.

---

### 3.2.2 INFERENCE WITH RENOISING

During inference, standard sampling algorithms can be applied within each pyramid stage. However, we must carefully handle the jump points (Campbell et al., 2023) between successive pyramid stages of different resolutions to ensure continuity of the probability path.

To ensure continuity, we first upsample the previous low-resolution endpoint with nearest or bilinear resampling. The result, as a linear combination of the input, follows a Gaussian distribution:

$$Up(\hat{\boldsymbol{x}}_{e_{k+1}})|\boldsymbol{x}_1 \sim \mathcal{N}(e_{k+1} \, Up(Down(\boldsymbol{x}_1, 2^{k+1})), (1 - e_{k+1})^2 \, \boldsymbol{\Sigma}), \tag{12}$$

where $\boldsymbol{\Sigma}$ is a covariance matrix depending on the upsampling function. Comparing Eqs. (8) and (12), we find it possible to match the Gaussian distributions at each jump point by a linear transformation of the upsampled result. Specifically, the following rescaling and renoising scheme would suffice:

$$\hat{\boldsymbol{x}}_{s_k} = \frac{s_k}{e_{k+1}} \, Up(\hat{\boldsymbol{x}}_{e_{k+1}}) + \alpha \boldsymbol{n}', \quad \text{s.t. } \boldsymbol{n}' \sim \mathcal{N}(\mathbf{0}, \boldsymbol{\Sigma}'), \tag{13}$$

where the rescaling coefficient $s_k/e_{k+1}$ allows matching the means of these distributions, and the corrective noise $\boldsymbol{n}'$ with a weight of $\alpha$ allows matching their covariance matrices.

To derive the corrective noise and its covariance, we consider a simplest scenario with nearest neighbor upsampling. In this case, $\boldsymbol{\Sigma}$ has a blockwise structure with non-zero elements only in the $4 \times 4$ blocks along the diagonal (corresponding to those upsampled from the same pixel). Then, it can be inferred that the corrective noise's covariance matrix $\boldsymbol{\Sigma}'$ also has a blockwise structure:

$$\boldsymbol{\Sigma}_{block} = \begin{pmatrix} 1 & 1 & 1 & 1 \\ 1 & 1 & 1 & 1 \\ 1 & 1 & 1 & 1 \\ 1 & 1 & 1 & 1 \end{pmatrix} \Rightarrow \boldsymbol{\Sigma}'_{block} = \begin{pmatrix} 1 & \gamma & \gamma & \gamma \\ \gamma & 1 & \gamma & \gamma \\ \gamma & \gamma & 1 & \gamma \\ \gamma & \gamma & \gamma & 1 \end{pmatrix}, \tag{14}$$

where $\boldsymbol{\Sigma}'_{block}$ contains negative elements $\gamma \in [-1/3, 0]$[1] to reduce the correlation within each block, as illustrated in Fig. 2b. Since it is desirable to maximally preserve the signals at each jump point, we opt to add a small amount of noise with $\gamma = -1/3$ such that it is most specialized for decorrelation. Substituting this into the above gives the update rule at jump points (see Appendix A for derivations):

$$\hat{\boldsymbol{x}}_{s_k} = \frac{1 + s_k}{2} \, Up(\hat{\boldsymbol{x}}_{e_{k+1}}) + \frac{\sqrt{3}(1 - s_k)}{2} \boldsymbol{n}', \tag{15}$$

with $e_{k+1} = 2s_k/(1 + s_k)$. The resulting inference process with renoising is shown in Algorithm 1.

### 3.3 PYRAMIDAL TEMPORAL CONDITION

Beyond the spatial complexity addressed in above sections, video presents another significant challenge due to its temporal length. The prevailing full-sequence diffusion methods generate all video frames simultaneously, restricting them to fixed-length generation (consistent with training). In contrast, the autoregressive video generation paradigm supports flexible-length generation during inference. Recent advancements (Chen et al., 2024a; Valevski et al., 2024) have also demonstrated its effectiveness in creating long-duration video content. However, their training is still severely limited by the computational complexity arising from the full-resolution long-history condition.

We observe that there is a high redundancy in full-resolution history conditions. For example, earlier frames in a video tend to provide high-level semantic conditions and are less related to appearance

---

[1]The lower bound $-1/3$ ensures that the covariance matrix is semidefinite.

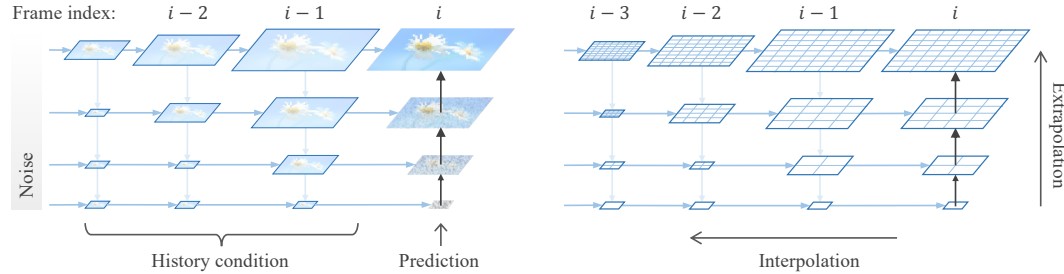

(a) Temporal pyramids rearranged in rows      (b) Position encoding in spatial and temporal pyramid

Figure 3: Illustration of temporal pyramid. (a) At each pyramid stage, the generation is conditioned on a compressed, lower-resolution history to improve training efficiency of the autoregressive model, as indicated by the rows. (b) A compatible position encoding scheme is devised that extrapolates in the spatial pyramid but interpolates in the temporal pyramid to allow spatial alignment of conditions.

details. This motivates us to use compressed, lower-resolution history for autoregressive video generation. As shown in Fig. 3a, we adopt a history condition of gradually increasing resolutions:

$$\underbrace{\ldots \to Down(\boldsymbol{x}_{t'}^{i-2}, 2^{k+1}) \to Down(\boldsymbol{x}_{t'}^{i-1}, 2^{k})}_{\text{History condition}} \to \underset{\underset{\text{Training}}{\uparrow}}{\hat{\boldsymbol{x}}_{t}^{i}} , \qquad (16)$$

where the superscripts are the history latent index, and the subscript $t'$ indicates small noise added to history latents in training to mitigate error accumulation with autoregressive generation, as in (Chen et al., 2024a; Valevski et al., 2024). After training, we use clean generated frames for inference:

$$\underbrace{\ldots \to Down(\boldsymbol{x}_{1}^{i-2}, 2^{k+1}) \to Down(\boldsymbol{x}_{1}^{i-1}, 2^{k})}_{\text{History condition}} \to \underset{\underset{\text{Prediction}}{\uparrow}}{\hat{\boldsymbol{x}}_{t}^{i}} . \qquad (17)$$

The above design significantly reduces the computational and memory overhead of video generative pre-training. Let there be $T$ history latents over $K$ lower resolutions, then most frames are computed at the lowest resolution of $1/2^{K}$, which reduces the number of training tokens by up to $1/4^{K}$ times. As a result, training efficiency is improved by up to $16^{K}/T$ times.

### 3.4 PRACTICAL IMPLEMENTATION

In this section, we show that the above pyramid designs can be easily implemented using standard Transformer architecture (Vaswani et al., 2017) and pipelines. This is crucial for efficient and scalable video generative pre-training based on existing acceleration frameworks.

Unlike previous methods (Ma et al., 2024) that utilize factorized spatial and temporal attention to reduce computational complexity, we directly employ full sequence attention, thanks to much fewer tokens required by our pyramidal representation. Furthermore, blockwise causal attention is adopted in each transformer layer, ensuring that each token cannot attend to its subsequent frames. The ablation results in Appendix C.2 illustrate that such casual attention design is crucial for autoregressive video generation. Another important design choice is the position encoding, as the pyramid designs introduce multiple spatial resolutions. As shown in Fig. 3b, we extrapolate position encoding in the spatial pyramid for better fine-grained detail (Yang et al., 2024), while interpolating it in the temporal pyramid input to spatially align the history conditions.

During training, different pyramidal stages are uniformly sampled in each update iteration. The autoregressive nature of our method inherently supports joint training of images and videos, since the first frame in a video acts as an image. We pack training samples with varying token counts together to form the length-balanced training batch following Patch n' Pack (Dehghani et al., 2023). After training, our method natively possesses the capability of text-to-video and text-conditioned image-to-video generation. During inference sampling, the classifier-free guidance strategy can be employed to enhance temporal consistency and motion smoothness of the generated video.

## 4 EXPERIMENTS

### 4.1 EXPERIMENTAL SETTINGS

**Training Dataset.** Our model is trained on a mixed corpus of open-source image and video datasets. For images, we utilize a high-aesthetic subset of LAION-5B (Schuhmann et al., 2022), 11M from CC-12M (Changpinyo et al., 2021), 6.9M non-blurred subset of SA-1B (Kirillov et al., 2023), 4.4M from JourneyDB (Sun et al., 2023), and 14M publicly available synthetic data. For video data, we incorporate the WebVid-10M (Bain et al., 2021), OpenVid-1M (Nan et al., 2024), and another 1M high-resolution non-watermark video primarily from the Open-Sora Plan (PKU-Yuan Lab et al., 2024). After postprocessing, around 10M single-shot videos are available for training.

**Evaluation Metrics.** We utilize the VBench (Huang et al., 2024) and EvalCrafter (Liu et al., 2024) for quantitative performance evaluation. VBench is a comprehensive benchmark that includes 16 fine-grained dimensions to systematically measure both motion quality and semantic alignment of video generative models. EvalCrafter is another large-scale evaluation benchmark including around 17 objective metrics for assessing video generation capabilities. In addition to automated evaluation metrics, we also conducted a study with human participants to measure the human preference for our generated videos. The compared baselines are summarized in Appendix B.

**Implementation Details.** We utilize the prevailing MM-DiT architecture from SD3 Medium (Esser et al., 2024) as the base model, with 2B parameters in total. It employs sinusoidal position encoding (Vaswani et al., 2017) in the spatial dimensions. As for the temporal dimension, the 1D Rotary Position Embedding (RoPE) (Su et al., 2024) is added to support flexible training with different video durations. In addition, we use a 3D Variational Autoencoder (VAE) to compress videos both spatially and temporally with a downsampling ratio of $8 \times 8 \times 8$. It shares a similar structure with MAGVIT-v2 (Yu et al., 2024) and is trained from scratch on the WebVid-10M dataset (Bain et al., 2021). The number of pyramid stages is set to 3 in all the experiments. Following Valevski et al. (2024), we add some corruptive noise of strength uniformly sampled from $[0, 1/3]$ to the history pyramid conditions, which is critical for mitigating the autoregressive generation degradation.

### 4.2 EFFICIENCY

The proposed pyramidal flow matching framework significantly reduces the computational and memory overhead in video generation training. Consider a video with $T$ frame latents, where each frame contains $N$ tokens at the original resolution. The full-sequence diffusion has $TN$ input tokens in DiT and requires $T^2N^2$ computations. In contrast, our method uses only approximately $TN/4^K$ tokens and $T^2N^2/16^K$ computations even for the final pyramid stage, which significantly improves the training efficiency. Specifically, it takes only 20.7k A100 GPU hours to train a 10s video generation model with 241 frames. Compared to existing models that require significant training resources, our method achieves superior video generation performance with much fewer computations. For example, the Open-Sora 1.2 (Zheng et al., 2024) requires 4.8k Ascend and 37.8k H100 hours to train the generation of only 97 video frames, consuming more than two times the computation of our approach, yet producing videos of worse quality. At inference, our model takes just 56 seconds to create a 5-second, 384p video clip, which is comparable to full-sequence diffusion counterparts.

### 4.3 MAIN RESULTS

**Text-to-Video Generation**. We first evaluate the text-to-video generation capability of the proposed method. For each text prompt, a 5-second 121 frames video is generated for evaluation. The detailed quantitative results on VBench (Huang et al., 2024) and EvalCrafter (Liu et al., 2024) are summarized in Tables 1 and 2, respectively. Overall, our method surpasses all the compared open-sourced video generation baselines in these two benchmarks. Even with only publicly accessible video data in training, it achieves comparable performance to commercial competitors trained on much larger proprietary data like Kling (Kuaishou, 2024) and Gen-3 Alpha (Runway, 2024). In particular, we demonstrated exceptional performance in quality score (84.74 vs. 84.11 of Gen-3), and motion smoothness in VBench, which are crucial criteria in reflecting the visual quality of generated videos. When evaluated in EvalCrafter, our method achieves better visual and motion quality scores than most compared methods. The semantic score is relatively lower than others, mainly because we use coarse-grained synthetic captions, which can be improved with more accurate video captioning.

Table 1: Experimental results on VBench (Huang et al., 2024). In terms of total score and quality score, our model even outperforms CogVideoX-5B (Yang et al., 2024) with twice the model size. In the following tables, we use blue to denote the highest scores among models trained on public data.

| Model | Public Data | Total Score | Quality Score | Semantic Score | Motion Smoothness | Dynamic Degree |
|---|---|---|---|---|---|---|
| Gen-2 | × | 80.58 | 82.47 | 73.03 | **99.58** | 18.89 |
| Pika 1.0 | × | 80.69 | 82.92 | 71.77 | 99.50 | 47.50 |
| CogVideoX-2B | × | 80.91 | 82.18 | 75.83 | 97.73 | 59.86 |
| CogVideoX-5B | × | 81.61 | 82.75 | **77.04** | 96.92 | **70.97** |
| Kling | × | 81.85 | 83.38 | 75.68 | 99.40 | 46.94 |
| Gen-3 Alpha | × | **82.32** | 84.11 | 75.17 | 99.23 | 60.14 |
| Open-Sora Plan v1.3 | ✓ | 77.23 | 80.14 | 65.62 | 99.05 | 30.28 |
| Open-Sora 1.2 | ✓ | 79.76 | 81.35 | 73.39 | 98.50 | 42.39 |
| VideoCrafter2 | ✓ | 80.44 | 82.20 | 73.42 | 97.73 | 42.50 |
| T2V-Turbo | ✓ | 81.01 | 82.57 | 74.76 | 97.34 | 49.17 |
| Ours | ✓ | 81.72 | **84.74** | 69.62 | 99.12 | 64.63 |

Table 2: Experimental results on EvalCrafter (Liu et al., 2024). See Appendix C.1 for raw metrics.

| Model | Public Data | Final Sum Score | Visual Quality | Text-Video Alignment | Motion Quality | Temporal Consistency |
|---|---|---|---|---|---|---|
| Pika 1.0 | × | 250 | 63.05 | 66.97 | **56.43** | 63.81 |
| Gen-2 | × | **254** | **69.09** | 63.92 | 55.59 | **65.40** |
| ModelScope | ✓ | 218 | 53.09 | 54.46 | 52.47 | 57.80 |
| Show-1 | ✓ | 229 | 52.19 | 62.07 | 53.74 | 60.83 |
| LaVie | ✓ | 234 | 57.99 | 68.49 | 52.83 | 54.23 |
| VideoCrafter2 | ✓ | 243 | 63.98 | 63.16 | 54.82 | 61.46 |
| Ours | ✓ | 244 | 67.94 | 57.01 | 55.31 | 63.41 |

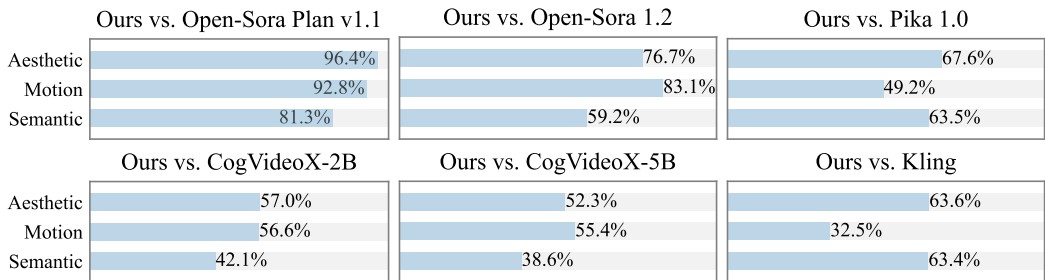

Figure 4: User preference on sampled VBench prompts. Our videos are generated at 5s, 768p, 24fps.

We also present some generated 5–10 second videos in Fig. 5, showing cinematic visual quality and validate the efficacy of pyramidal flow matching. More visualizations are provided in Appendix C.3.

**User study**. While quantitative evaluation scores reflect the video generation capability to some extent, they may not align with human preferences for visual quality. Hence, an additional user study is conducted to compare our performance with six baseline models, including CogVideoX (Yang et al., 2024) and Kling (Kuaishou, 2024). We utilized 50 prompts sampled from VBench and asked 20+ participants to rank each model according to the aesthetic quality, motion smoothness, and semantic alignment of the generated videos. As seen in Fig. 4, our method is preferred over open-source models such as Open-Sora and CogVideoX-2B especially in terms of motion smoothness. This is due to the substantial token savings achieved by pyramidal flow matching, enabling generation of 5-second (up to 10-second) 768p videos at 24 fps, while the baselines usually support video synthesis of similar length only at 8 fps. The detailed user study settings are presented in Appendix B.

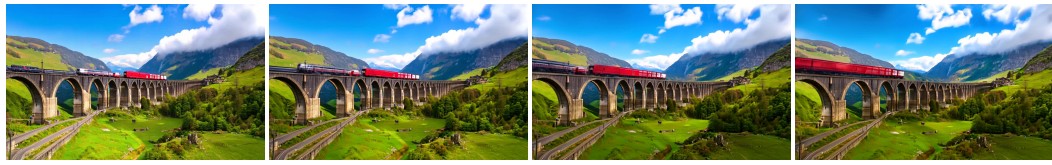

(a) The Glenfinnan Viaduct is a historic railway bridge... It is a stunning sight as a steam train leaves the bridge, traveling over the arch-covered viaduct. The landscape is dotted with lush greenery and rocky mountains...

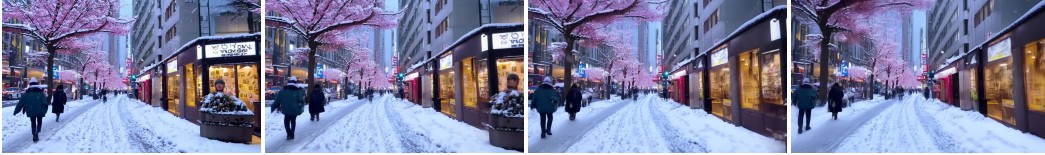

(b) Beautiful, snowy Tokyo city is bustling. The camera moves through the bustling city street, following several people enjoying the beautiful snowy weather and shopping at nearby stalls. Gorgeous sakura petals...

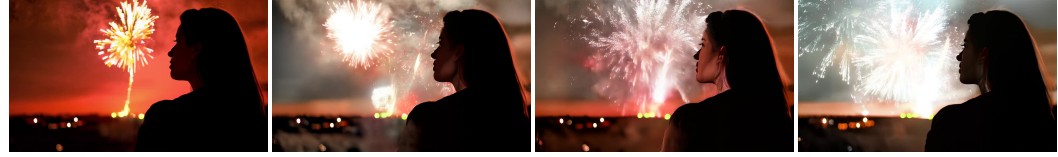

(c) A side profile shot of a woman with fireworks exploding in the distance beyond her.

Figure 5: Visualization of text-to-video generation results. The top two videos are generated at 5s, 768p, 24fps, and the bottom one at 10s, 768p, 24fps. See more generated videos on our project page.

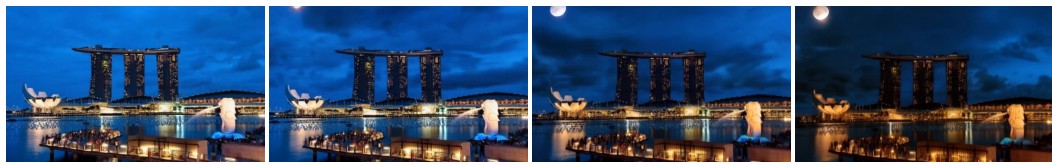

(a) A moon rises from the sky and the lights on the land are bright.

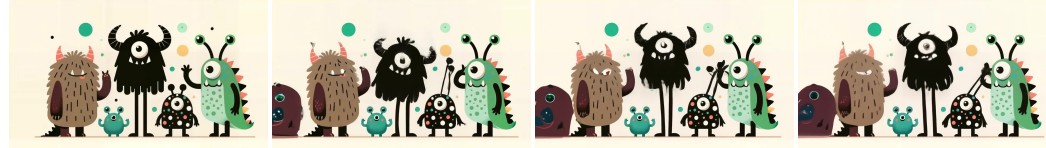

(b) Monster Illustration in flat design style of a diverse family of monsters. The group includes a furry brown monster, a sleek black monster with antennas, a spotted green monster, and a tiny polka-dotted monster, all...

Figure 6: Visualization of text-conditioned image-to-video generation results (5s, 768p, 24fps).

**Image-to-Video Generatetion**. Thanks to the autoregressive property of our model and the causal attention design, the first frame of each video acts similarly to an image condition during the training. Consequently, although our model is optimized solely for text-to-video generation, it naturally accommodates text-conditioned image-to-video generation during inference. Given an image and a textual prompt, it is able to animate the static input image by autoregressively predicting the future frames without further fine-tuning. In Fig. 6, we illustrate qualitative examples of its image-to-video generation performance, where each example consists of 120 newly synthesized frames spanning a duration of 5 seconds. As can be seen, our model successfully predicts reasonable subsequent motion, endowing the images with rich temporal dynamic information. More generated video examples are best viewed on our project page at `https://pyramid-flow.github.io`.

## 4.4 ABLATION STUDY

In this section, we conduct ablation studies to validate the crucial component of our methods, including the spatial pyramid in denoising trajectory and the temporal pyramid in history condition. Due to limited space, the ablations for other design choices are provided in Appendix C.2.

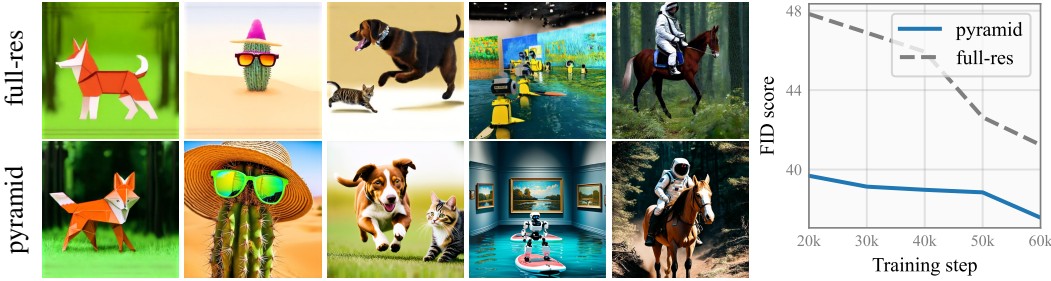

Figure 7: Ablation study of spatial pyramid at 50k image training step. On the right is a quantitative comparison of the FID results, where our method achieves almost three times the convergence speed.

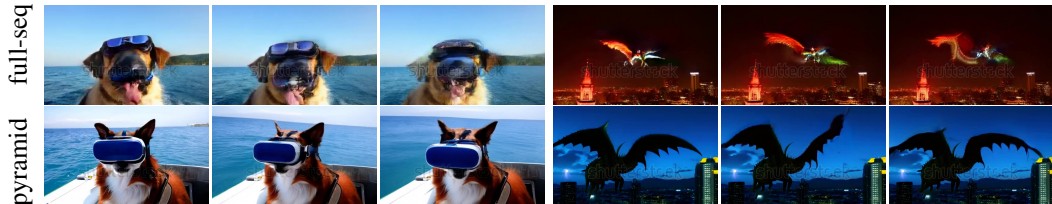

Figure 8: Ablation study of temporal pyramid at 100k low-resolution video training step.

**Effectiveness of spatial pyramid.** In the generation trajectory of the proposed spatial pyramid, only the final stage operates at full resolution, which significantly reduces the number of tokens for most denoising timesteps. With the same computational resources, it can handle more samples per training batch, greatly enhancing the convergence rate. To validate its efficiency, we designed a baseline that employs the standard flow matching objective for training text-to-image generation in our early experiments. This baseline is optimized using the same training data, number of tokens per batch, hyperparameter configurations, and model architecture to ensure fairness. The performance comparison is illustrated in Fig. 7. It can be observed that the variant using pyramidal flow demonstrates superior visual quality and prompt-following capability. We further quantitatively evaluate the FID metric of these methods on the MS-COCO benchmark (Lin et al., 2014) by randomly sampling 3K prompts. The FID performance curve over training steps is presented on the right of Fig. 7. Compared to standard flow matching, the convergence rate of our method is significantly improved.

**Effectiveness of temporal pyramid.** As mentioned in Section 4.2, the temporal pyramid design can drastically reduce the computation demands compared to traditional full-sequence diffusion. Similar to the spatial pyramid, we also established a full-sequence diffusion baseline under the same experimental setting to investigate its training efficiency improvement. The qualitative comparison with the baseline is presented in Fig. 8, where the generated videos of our pyramidal variant demonstrate much better visual quality and temporal consistency under the same training steps. In contrast, the full-sequence diffusion baseline is far from convergence. It fails to produce coherent motion, leading to fragmented visual details and severe artifacts in the generated videos. This performance gap clearly highlights the training acceleration achieved by our method in video generative modeling.

## 5 CONCLUSION

This work presents an efficient video generative modeling framework based on pyramidal visual representations. In contrast to cascaded diffusion models that use separate models for different image pyramids to improve efficiency, we propose a unified pyramidal flow matching objective that simultaneously generates and decompresses visual content across pyramid stages with a single model, effectively facilitating knowledge sharing. Furthermore, a temporal pyramid design is introduced to reduce computational redundancy in the full-resolution history of a video. The proposed method is extensively evaluated on VBench and EvalCrafter, demonstrating advantageous performance.

**Reproducibility Statement.** Our code and models are open-sourced at `https://pyramid-flow.github.io`. The experimental settings are detailed in Section 4.1 and Appendix B.

ACKNOWLEDGMENTS

The work was supported by National Key R&D Program of China (2022ZD0160300), an internal grant of Peking University (2024JK28), a grant from Kuaishou (No. DJHL-20240809-115) and NSF China (No. 62276004).

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

## A DERIVATION

This section provides a detailed derivation for Eq. (15) that handles jump points in the spatial pyramid. For quick lookup, Table 3 summarizes the used notations.

| Symbol | Description |
|---|---|
| $\boldsymbol{x}_1$ | Data latent at full resolution |
| $\boldsymbol{x}_0$ | Noise at full resolution |
| $K$ | Number of pyramid stages |
| $e_k$ | Timestep at endpoint of $k$-th pyramid stage |
| $s_k$ | Timestep at starting point of $k$-th pyramid stage |
| $\hat{\boldsymbol{x}}_{e_k}$ | Noisy latent at endpoint of $k$-th stage |
| $\hat{\boldsymbol{x}}_{s_k}$ | Noisy latent at starting point of $k$-th stage |
| $\hat{\boldsymbol{x}}_t$ | Noisy latent at timestep $t$ |
| $\boldsymbol{n}$ | Noise at the resolution of current stage |
| $Up(\cdot)$ | Upsampling function, *e.g.* nearest-neighbor |
| $Down(\cdot, \cdot)$ | Downsampling function, *e.g.* bilinear |

Table 3: Notation in the main paper.

To ensure continuity of the probability path across different stages of the spatial pyramid, we need to make sure that the endpoints have the same probability distribution. According to Eqs. (8) and (12), their distributions are already similar after a simple upsampling transformation:

$$\hat{\boldsymbol{x}}_{s_k}|\boldsymbol{x}_1 \sim \mathcal{N}(s_k\, Up(Down(\boldsymbol{x}_1, 2^{k+1})), (1-s_k)^2\boldsymbol{I}), \tag{18}$$

$$Up(\hat{\boldsymbol{x}}_{e_{k+1}})|\boldsymbol{x}_1 \sim \mathcal{N}(e_{k+1}\, Up(Down(\boldsymbol{x}_1, 2^{k+1})), (1-e_{k+1})^2\,\boldsymbol{\Sigma}). \tag{19}$$

Therefore, we can directly apply a linear transformation with a corrective Gaussian noise to match their distributions:

$$\hat{\boldsymbol{x}}_{s_k} = \frac{s_k}{e_{k+1}}\, Up(\hat{\boldsymbol{x}}_{e_{k+1}}) + \alpha\boldsymbol{n}', \quad \text{s.t. } \boldsymbol{n}' \sim \mathcal{N}(\boldsymbol{0}, \boldsymbol{\Sigma}'), \tag{20}$$

where the rescaling coefficient $s_k/e_{k+1}$ allows the means of these distributions to be matched, and $\alpha$ is the noise weight. Additionally, we need to match the covariance matrices of Eqs. (18) and (20):

$$\frac{s_k^2}{e_{k+1}^2}(1-e_{k+1})^2\boldsymbol{\Sigma} + \alpha^2\boldsymbol{\Sigma}' = (1-s_k)^2\boldsymbol{I}. \tag{21}$$

To allow analysis of covariance matrices, *e.g.* $\boldsymbol{\Sigma}$, we consider a simplest scenario with nearest neighbor upsampling. In this case, $\boldsymbol{\Sigma}$ has a blockwise structure with non-zero elements only in the $4 \times 4$ blocks along the diagonal (corresponding to those upsampled from the same pixel). Then, it can be inferred that the corrective noise's covariance matrix $\boldsymbol{\Sigma}'$ has a similar blockwise structure:

$$\boldsymbol{\Sigma}_{block} = \begin{pmatrix} 1 & 1 & 1 & 1 \\ 1 & 1 & 1 & 1 \\ 1 & 1 & 1 & 1 \\ 1 & 1 & 1 & 1 \end{pmatrix} \Rightarrow \boldsymbol{\Sigma}'_{block} = \begin{pmatrix} 1 & \gamma & \gamma & \gamma \\ \gamma & 1 & \gamma & \gamma \\ \gamma & \gamma & 1 & \gamma \\ \gamma & \gamma & \gamma & 1 \end{pmatrix}, \tag{22}$$

where $\gamma$ is a negative value in $[-1/3, 0]$ for the decorrelation (its lower bound $-1/3$ ensures that the covariance matrix is semidefinite). We further rewrite Eqs. (21) and (22) by considering the equality of their diagonal and non-diagonal elements, respectively:

$$\frac{s_k^2}{e_{k+1}^2}(1-e_{k+1})^2 + \alpha^2 = (1-s_k)^2, \tag{23}$$

$$\frac{s_k^2}{e_{k+1}^2}(1-e_{k+1})^2 + \alpha^2\gamma = 0. \tag{24}$$

Taking into account the timestep constraints $0 < s_k, e_{k+1} < 1$, they can be solved directly:

$$e_{k+1} = \frac{s_k\sqrt{1-\gamma}}{(1-s_k)\sqrt{-\gamma} + s_k\sqrt{1-\gamma}}, \quad \alpha = \frac{1-s_k}{\sqrt{1-\gamma}}. \tag{25}$$

Intuitively, it is desirable to maximally preserve the signals at each jump point, which corresponds to minimizing the noise weight $\alpha$. According to Eq. (25), this is equivalent to minimizing $\gamma$. Substituting its minimum value $\gamma = -1/3$ into Eq. (25) yields:

$$e_{k+1} = \frac{2s_k}{1 + s_k}, \quad \alpha = \frac{\sqrt{3}(1 - s_k)}{2}. \tag{26}$$

It is worth noting that $e_{k+1} > s_k$, indicating that the timestep is rolled back a bit when adding the corrective noise at each jump point. We can further obtain the renoising rule in Eq. (15):

$$\hat{\boldsymbol{x}}_{s_k} = \frac{1 + s_k}{2} \, Up(\hat{\boldsymbol{x}}_{e_{k+1}}) + \frac{\sqrt{3}(1 - s_k)}{2} \boldsymbol{n}'. \tag{27}$$

## B  EXPERIMENTAL SETTINGS

**Model Implementation Details.** We adopt the MM-DiT architecture, based on SD3 Medium (Esser et al., 2024), which comprises 24 transformer layers and a total of 2B parameters. The weights of the MM-DiT are initialized from the SD3 medium. Following the more recent FLUX.1 (Black Forest Labs, 2024), both T5 (Raffel et al., 2020) and CLIP (Radford et al., 2021) encoders are employed for prompts embedding. To address the redundancy in video data, we have designed a 3D VAE that compresses videos both spatially and temporally into a latent space. The architecture of this VAE is similar to MAGVIT-v2 (Yu et al., 2024), employing 3D causal convolution to ensure that each frame depends only on the preceding frames. It features an asymmetric encoder-decoder with Kullback-Leibler (KL) regularization applied to the latents. Overall, the 3D VAE achieves a compression rate of $8 \times 8 \times 8$ from pixels to the latent. It is trained on WebVid-10M and 6.9M SAM images from scratch. To support the tokenization of very long videos, we scatter them into multiple GPUs to distribute computation like CogVideoX (Yang et al., 2024).

**Training Procedure** Our model undergoes a three-stage training procedure using 128 NVIDIA A100 GPUs. (1) Image Training. In the first stage, we utilize a pure image dataset that includes 180M images from LAION-5B (Schuhmann et al., 2022), 11M from CC-12M (Changpinyo et al., 2021), 6.9M non-blurred images from SA-1B (Kirillov et al., 2023), and 4.4M from JourneyDB (Sun et al., 2023). We keep the image's original aspect ratio and rearrange them into different buckets. It is trained for a total of 50,000 steps, requiring approximately 1536 A100 GPU hours. After this stage, the model has learned the dependencies between visual pixels, which facilitates the convergence of subsequent video training. (2) Low-Resolution Video Training. For this stage, we employ the WebVid-10M (Bain et al., 2021), OpenVid-1M (Nan et al., 2024), and another 1M non-watermark video from the Open-Sora Plan (PKU-Yuan Lab et al., 2024). We also leverage the Video-LLaMA2 (Cheng et al., 2024), a state-of-the-art video understanding model, to recaption each video sample. The image data from stage 1 is also utilized at a proportion of 12.5% in each batch. We first train the model for 80,000 steps on 2-second video generation, followed by an additional 120,000 steps on 5-second videos. In total, it takes about 11,520 A100 GPU hours at this stage. (3) High-Resolution Video Training. The final stage employs the same strategy to continue fine-tuning the model on the aforementioned high-resolution video dataset of varying durations (5–10s). It consumes approximately 7,680 A100 GPU hours for 50,000 steps in the final stage.

**Hyperparameters Setting** The detailed training hyper-parameter settings for each optimization stage are reported in Table 4.

**Baseline Methods.** For VBench (Huang et al., 2024), we compare with eight baseline methods, including Open-Sora Plan V1.3 (PKU-Yuan Lab et al., 2024), Open-Sora 1.2 (Zheng et al., 2024), VideoCrafter2 (Chen et al., 2024b), Gen-2 (Runway, 2023), Pika 1.0 (Pika, 2023), T2V-Turbo (Li et al., 2024), CogVideoX (Yang et al., 2024), Kling (Kuaishou, 2024), and Gen-3 Alpha (Runway, 2024). Among them, Open-Sora Plan, Open-Sora, CogVideo-X, Kling and Gen-3 Alpha can generate long videos. For EvalCrafter (Liu et al., 2024), our model is compared to six baselines, including ModelScope (Wang et al., 2023a), Show-1 (Zhang et al., 2023a), LaVie (Wang et al., 2023b), VideoCrafter2 (Chen et al., 2024b), Pika 1.0 (Pika, 2023), and Gen-2 (Runway, 2023). The above models are all based on full-sequence diffusion, while our method combines the merits of autoregressive generation and flow generative models to achieve better training efficiency of video generation.

**User Study.** To complement the quantitative evaluation in the main paper, we conduct a rigorous user study to collect human preferences for these generative models. To accomplish this, we sample

| Configuration | Stage-1 | Stage-2 | Stage-3 |
|---|---|---|---|
| Optimizer | AdamW | AdamW | AdamW |
| Optimizer Hyperparameters | $\beta_1 = 0.9, \beta_2 = 0.999, \epsilon = 1e^{-6}$ | $\beta_1 = 0.9, \beta_2 = 0.95, \epsilon = 1e^{-6}$ | |
| Global batch size | 1536 | 768 | 384 |
| Learning rate | 1e-4 | 1e-4 | 5e-5 |
| Learning rate schedule | Constant with warmup | Constant with warmup | Constant with warmup |
| Training Steps | 50k | 200k | 50k |
| Warm-up steps | 1k | 1k | 1k |
| Weight decay | 1e-4 | 1e-4 | 1e-4 |
| Gradient clipping | 1.0 | 1.0 | 1.0 |
| Numerical precision | bfloat16 | bfloat16 | bfloat16 |
| GPU Usage | 128 NVIDIA A100 | 128 NVIDIA A100 | 128 NVIDIA A100 |
| Training Time | 12h | 90h | 60h |

Table 4: The detailed training hyperparameters of our method

Figure 9: Interface for user study of video generative performance.

50 prompts from the VBench prompt list and randomly sample one generated video for each prompt from the baseline model. In total, six baseline models are considered, including Open-Sora Plan V1.1 (PKU-Yuan Lab et al., 2024), Open-Sora 1.2 (Zheng et al., 2024), Pika 1.0 (Pika, 2023), CogVideoX-2B and 5B (Yang et al., 2024), and Kling (Kuaishou, 2024). We then pair these results with our generated video and ask the participant to rank their preference among three dimensions: aesthetic quality, motion smoothness, and semantic alignment, each of which represents a crucial aspect of video quality. The interface for the user study is exemplified in Fig. 9, where the user accepts a prompt and two generated videos (with the unnecessary information cropped, such as a watermark indicating which model it belongs to), and chooses between which model is better in the three dimensions. We distribute the user study to more than 20 participants, and collect a total of 1411 valid preference choices, ensuring its effectiveness. The results of this user study are presented in Fig. 4, where our model shows a very competitive performance among the compared baselines.

## C ADDITIONAL RESULTS

### C.1 QUANTITATIVE RESULTS

This section provides the full results on VBench (Huang et al., 2024) and EvalCrafter (Liu et al., 2024) as a supplement to the performance comparison in the experiments section of the main paper. The evaluation of our model is performed using 5-second 768p videos generated at 24 fps.

Table 5: Detailed results on VBench (Huang et al., 2024). See Table 1 for the summarized results. We additionally use blue to indicate the highest scores among models trained on public datasets.

| Model | Subject Consistency | Background Consistency | Temporal Flickering | Motion Smoothness | Dynamic Degree | Aesthetic Quality | Imaging Quality | Object Class |
|---|---|---|---|---|---|---|---|---|
| *Trained on private datasets:* | | | | | | | | |
| Gen-2 | 97.61 | 97.61 | 99.56 | **99.58** | 18.89 | **66.96** | 67.42 | 90.92 |
| Pika 1.0 | 96.94 | 97.36 | **99.74** | 99.50 | 47.50 | 62.04 | 61.87 | 88.72 |
| CogVideoX-2B | 96.78 | 96.63 | 98.89 | 97.73 | 59.86 | 60.82 | 61.68 | 83.37 |
| CogVideoX-5B | 96.23 | 96.52 | 98.66 | 96.92 | **70.97** | 61.98 | 62.90 | 85.23 |
| Kling | **98.33** | 97.60 | 99.30 | 99.40 | 46.94 | 61.21 | 65.62 | 87.24 |
| Gen-3 Alpha | 97.10 | 96.62 | 98.61 | 99.23 | 60.14 | 63.34 | 66.82 | 87.81 |
| *Trained on public datasets:* | | | | | | | | |
| Open-Sora Plan v1.3 | 97.79 | 97.24 | 99.20 | 99.05 | 30.28 | 60.42 | 56.21 | 85.56 |
| Open-Sora 1.2 | 96.75 | 97.61 | 99.53 | 98.50 | 42.39 | 56.85 | 63.34 | 82.22 |
| VideoCrafter2 | 96.85 | 98.22 | 98.41 | 97.73 | 42.50 | 63.13 | 67.22 | 92.55 |
| T2V-Turbo | 96.28 | 97.02 | 97.48 | 97.34 | 49.17 | 63.04 | 72.49 | 93.96 |
| Ours | 96.95 | 98.06 | 99.49 | 99.12 | 64.63 | 63.26 | 65.01 | 86.67 |

| Model | Multiple Objects | Human Action | Color | Spatial Relationship | Scene | Appearance Style | Temporal Style | Overall Consistency |
|---|---|---|---|---|---|---|---|---|
| *Trained on private datasets:* | | | | | | | | |
| Gen-2 | 55.47 | 89.2 | 89.49 | 66.91 | 48.91 | 19.34 | 24.12 | 26.17 |
| Pika 1.0 | 43.08 | 86.2 | 90.57 | 61.03 | 49.83 | 22.26 | 24.22 | 25.94 |
| CogVideoX-2B | 62.63 | 98.0 | 79.41 | 69.90 | 51.14 | 24.80 | 24.36 | 26.66 |
| CogVideoX-5B | 62.11 | **99.4** | 82.81 | 66.35 | 53.20 | 24.91 | 25.38 | 27.59 |
| Kling | **68.05** | 93.4 | 89.90 | **73.03** | 50.86 | 19.62 | 24.17 | 26.42 |
| Gen-3 Alpha | 53.64 | 96.4 | 80.90 | 65.09 | 54.57 | 24.31 | 24.71 | 26.69 |
| *Trained on public datasets:* | | | | | | | | |
| Open-Sora Plan v1.3 | 43.58 | 86.8 | 79.30 | 51.61 | 36.73 | 20.03 | 22.47 | 24.47 |
| Open-Sora 1.2 | 51.83 | 91.2 | 90.08 | 68.56 | 42.44 | 23.95 | 24.54 | 26.85 |
| VideoCrafter2 | 40.66 | 95.0 | 92.92 | 35.86 | 55.29 | 25.13 | 25.84 | 28.23 |
| T2V-Turbo | 54.65 | 95.2 | 89.90 | 38.67 | 55.58 | 24.42 | 25.51 | 28.16 |
| Ours | 50.71 | 85.6 | 82.87 | 59.53 | 43.20 | 20.91 | 23.09 | 26.23 |

Table 6: Raw metrics on EvalCrafter (Liu et al., 2024). The baseline results are found on its website, but there were no results for LaVie (Wang et al., 2023b). See Table 2 for the summarized results.

| Model | VQA$_A$ | VQA$_T$ | IS | CLIP-Temp | Warping Error | Face Consistency | Action-Score | Motion AC-Score |
|---|---|---|---|---|---|---|---|---|
| *Trained on private datasets:* | | | | | | | | |
| Pika 1.0 | 69.23 | 71.12 | 16.67 | 99.89 | 0.0008 | 99.22 | 61.29 | 42.0 |
| Gen-2 | **90.39** | **92.18** | **19.28** | **99.99** | **0.0005** | 99.35 | 73.44 | 44.0 |
| *Trained on public datasets:* | | | | | | | | |
| ModelScope | 40.06 | 32.93 | 17.64 | 99.74 | 0.0162 | 98.94 | 72.12 | 42.0 |
| Show-1 | 23.19 | 44.24 | 17.65 | 99.77 | 0.0067 | 99.32 | **81.56** | **50.0** |
| VideoCrafter2 | 79.93 | 67.04 | 17.39 | 99.84 | 0.0085 | **99.44** | 68.17 | 36.0 |
| Ours | 86.09 | 88.31 | 18.49 | 99.90 | 0.0019 | 98.89 | 67.58 | 46.0 |

| Model | Flow-Score | CLIP-Score | BLIP-BLUE | SD-Score | Detection-Score | Color-Score | Count-Score | OCR-Score | Celebrity ID Score |
|---|---|---|---|---|---|---|---|---|---|
| *Trained on private datasets:* | | | | | | | | | |
| Pika 1.0 | 1.14 | 20.47 | 21.31 | 67.43 | **70.26** | 42.03 | **62.19** | 94.85 | **36.53** |
| Gen-2 | 0.58 | 20.26 | 22.25 | 67.69 | 69.54 | 47.39 | 58.36 | 63.74 | 38.90 |
| *Trained on public datasets:* | | | | | | | | | |
| ModelScope | 6.99 | 20.36 | 22.54 | 67.93 | 50.01 | 38.72 | 44.18 | 71.32 | 44.56 |
| Show-1 | 2.07 | 20.66 | 23.24 | 68.42 | 58.63 | 48.55 | 44.31 | 58.97 | 37.93 |
| VideoCrafter2 | 3.90 | 21.21 | 22.71 | 68.58 | 69.32 | 45.11 | 50.45 | 80.37 | 38.40 |
| Ours | 1.79 | 20.73 | 23.29 | 68.26 | 69.55 | 47.74 | 56.31 | 68.55 | 44.72 |

**VBench** (Huang et al., 2024). The full experimental results on VBench are shown in Table 5. As can be observed, our model achieves leading or highly competitive results among open-source and commercial competitors, especially for the metrics related to motion quality. For example, the dynamic

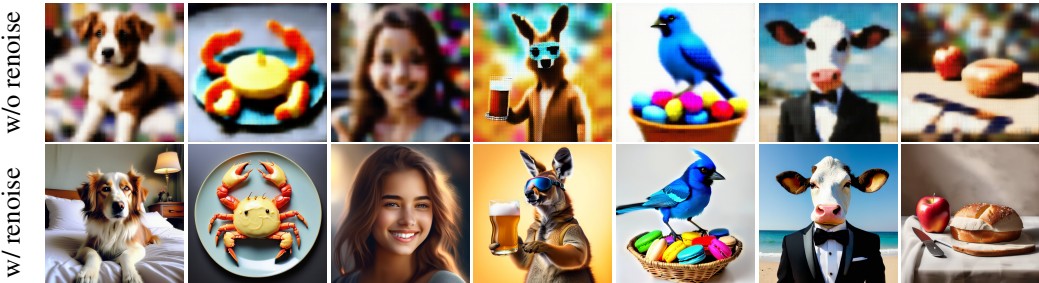

Figure 10: Ablation study of corrective renoising during the inference stage.

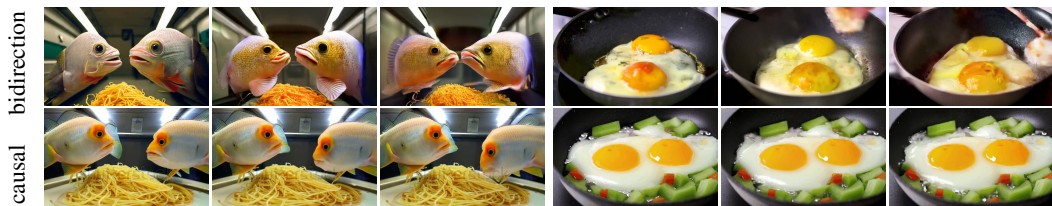

Figure 11: Ablation study of blockwise causal attention at 100k training step.

degree metric of our model ranks 2nd among all models at 64.63, validating the effectiveness of our generative model in learning temporal dynamics. For the rest of the metrics, our results are also generally superior to the open-source Open-Sora Plan v1.3 (PKU-Yuan Lab et al., 2024) and Open-Sora 1.2 (Zheng et al., 2024), with significantly lower training computational cost as mentioned earlier. We also note that half of our results even outperformed the recent CogVideoX-5B (Yang et al., 2024), which is based on a larger DiT model, demonstrating its modeling capacity. On the other hand, our model performs relatively inferior on metrics such as color and appearance style, which is more related to the image generation capabilities and finer-grained prompt following. **This is largely due to our video captioning procedure based on video LLMs which tends to produce coarse-grained captions, thus dampening these abilities. Nevertheless, thanks to our autoregressive generation framework, which decomposes video generation into first frame generation and subsequent frame generation, these image quality issues can be addressed separately with additional well-captioned image data in future training stages.** Similarly, due to the SD3-Medium weight initialization, which is infamous for its human structure, our method achieves a relatively low score in human action, which could be addressed by switching to other base models or training from scratch.

**EvalCrafter** (Liu et al., 2024). The raw metrics on EvalCrafter are provided in Table 6. Overall, our model delivers highly competitive performance on the majority of metrics, outperforming many previous open-source and closed-source models. In particular, the motion AC score of our method which is relevant to the temporal motion quality ranks 2nd among all methods, justifying the capacity of our pyramid designs to learn complex spatiotemporal patterns in video. Our method also demonstrates superiority over several other metrics related to semantic alignment, including BLIP-BLUE and CLIP score. Placing top two in both metrics among the models compared, including the closed-source Gen-2 (Runway, 2023), confirms the advantages of our model in text-to-video semantic alignment. The only metric where our model performs poorly is face consistency, which is due to the temporal pyramid design adopted for compressing the history condition. We view this as an issue that can potentially be addressed by better temporal compression schemes.

## C.2    ABALTION STUDY

In this section, we conduct additional ablation studies of two important design details in our proposed pyramidal flow matching, including the corrective noise added during inference of the spatial pyramid and the blockwise causal attention used for autoregressive video generation.

**Role of corrective noise.** To study its efficacy in the spatial pyramid, we curate a baseline method that inferences without adding this corrective Gaussian noise. The detailed comparative results

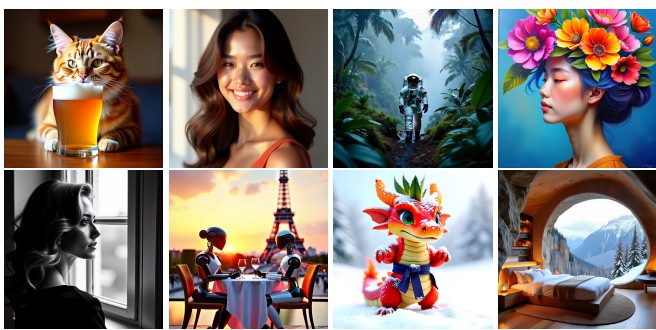
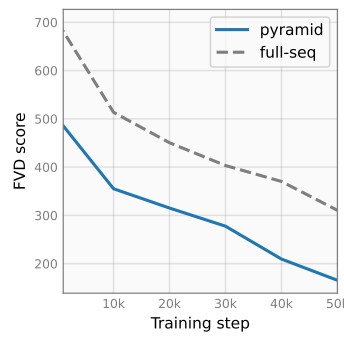

<p style="text-align:center">(a) Text-to-Image generation results.      (b) FVD convergence plot.</p>

Figure 12: (a) The visualization of generated images from our pyramid-flow. Our model can synthesize high-resolution and good-quality images even using only a few million training samples. (b) The FVD score comparison with full-sequence diffusion video training on MSR-VTT (Xu et al., 2016) benchmark along with optimization iterations.

of our method against this variant are shown in Fig. 10. While the baseline method has a correct global structure, it fails to produce a fine-grained, high-resolution image with rich details and instead produces a blurred image that suffers from block-like artifacts (better observed when zooming in). This is because applying the upsampling function at the jump points between different pyramid stages of varying resolutions results in excessive correlation between spatially adjacent latent values. In comparison, our generated images have rich details and vivid colors, confirming that the adopted corrective renoising scheme effectively addresses this artifact problem in the spatial pyramid.

**Effectiveness of causal attention.** In Fig. 11, we study the effect of blockwise causal attention by comparing it to the bidirectional attention used in full-sequence diffusion. While an intuitive understanding might be that bidirectional attention promotes information exchange and increases model capacity, it is understudied for autoregressive video generation. In an early experiment, we trained a baseline model using bidirectional attention across different latent frames, the results of which are visualized in Fig. 11. As can be seen from the sampled keyframes of the 1-second videos, this model suffers from a lack of temporal coherence as the subject in the generated video is constantly changing in shape and color. Meanwhile, our model shows good temporal coherence with reasonable motion. We infer that this is because the history condition in bidirectional attention is influenced by the ongoing generation and thus deviates, whereas the history condition in causal attention is fixed, serving as a predetermined condition and stabilizing the autoregressive generative process.

### C.3 VISUALIZATION

This section presents additional qualitative results for our text-to-video generation in comparison to the recent leading models including Gen-3 Alpha (Runway, 2024), Kling (Kuaishou, 2024) and CogVideoX (Yang et al., 2024). The uniformly sampled frames from the generated videos are shown in Figs. 14 and 15, in which our videos are generated at 5s, 768p, 24fps. Overall, we observe that despite being trained only on publicly available data and using a small computational budget, our model yields a highly competitive visual aesthetics and motion quality among the baselines.

Specifically, the results highlight the following characteristics of our model: (1) Through generative pre-training, our model is capable of generating videos of cinematic quality and reasonable content. For example, in Fig. 14a, our generative video shows a mushroom cloud resulting from "a massive explosion" taking place in "the surface of the earth", creating a sci-fi movie atmosphere. However, the current model is not fully faithful to some prompts such as the "salt desert" in Fig. 14b, which could be addressed by curating more high-quality caption data. (2) Despite that our model has only 2B parameters initialized from SD3-Medium (Esser et al., 2024), it clearly outperforms CogVideoX-2B of the same model size with additional training data, and is even comparable to the 5B full version in some aspects. For example, in Figs. 15a and 15b, only our model and its 5B version are capable of generating reasonable sea waves according to the input prompt, while its 2B variant merely illustrates an almost static sea surface. This is largely attributed to our proposed pyramidal

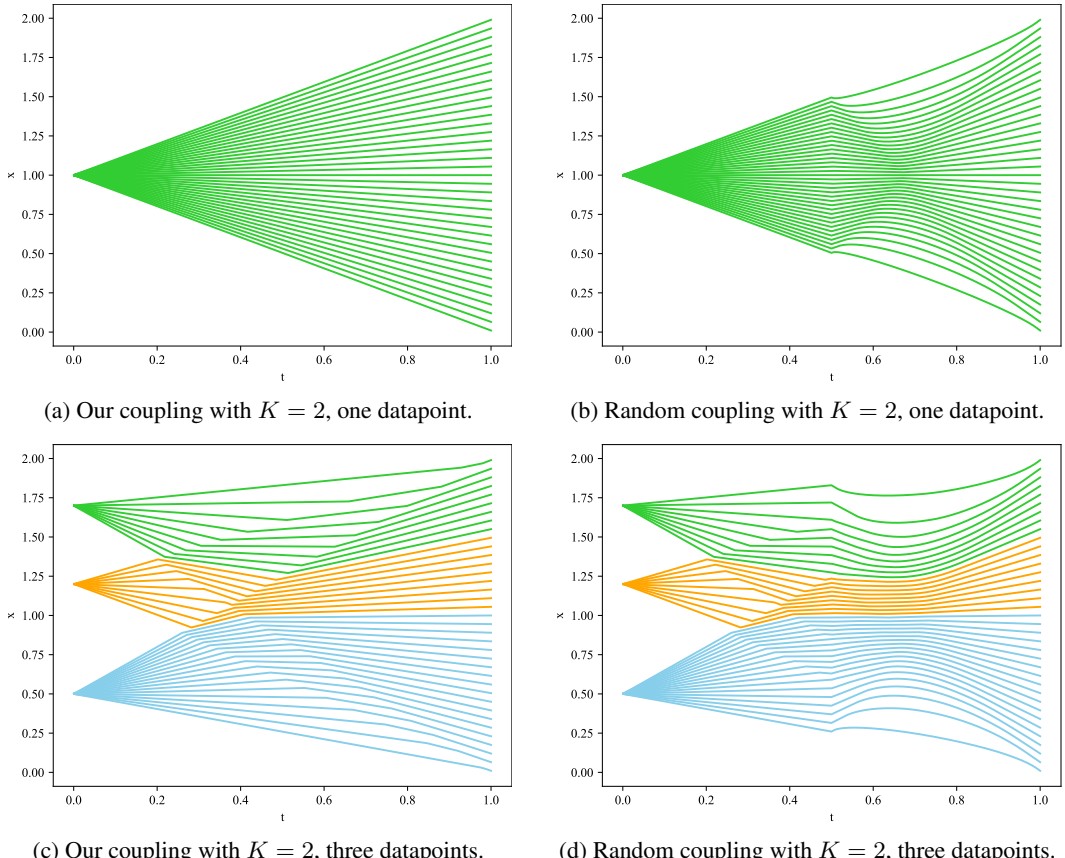

(a) Our coupling with $K = 2$, one datapoint.

(b) Random coupling with $K = 2$, one datapoint.

(c) Our coupling with $K = 2$, three datapoints.

(d) Random coupling with $K = 2$, three datapoints.

Figure 13: Justification for our coupled sampling in Eqs. (9) and (10).

flow matching in improving training efficiency. Overall, these results validate the effectiveness of our approach in modeling complex spatiotemporal patterns through the spatial and temporal pyramid designs. Our generated videos are best-viewed at `https://pyramid-flow.github.io`. Since the autoregressive video generation model natively generates a high-quality image as the first frame, pyramid-flow can also be applied to text-to-image generation. Even with only a few million training images, it can show excellent visual quality, see Fig. 12a for the generated images.

### C.4 TOY EXPERIMENT OF COUPLING NOISE

To validate the effectiveness of coupled sampling in Eqs. (9) and (10), we illustrate two variants of piecewise flow matching in a toy experiment that considers mapping a few data points to uniform distribution. Two different coupling designs are considered within each time window, namely our coupling *vs.* random coupling. It can be seen that our coupled sampling strategy produces much more straight flow trajectories.

The rationale for improving straightness by coupling noise is that: the straightness of the flow trajectory is usually compromised when there are intersections. Sampling the endpoints independently (as in vanilla flow matching) creates random directions for each trajectory and leads to intersections. Instead, by coupling the sampling of these endpoints as in Eq. (9) and Eq. (10), we can create more organized, possibly parallel trajectories with fewer intersections, thus improving straightness. As illustrated in Fig. 13, where coupling noise indeed leads to more straight flow trajectories.

### D LIMITATIONS

Our method only supports autoregressive generation and cannot be extended to keyframe interpolation or video interpolation. In addition, we noticed that the temporal pyramid designs to improve

training efficiency can sometimes lead to subtle subject inconsistency, especially over the long term. While this is not a prevalent problem, we believe that developing better temporal compression methods is critical to the broader applicability of autoregressive video generative model. In addition, improving inference efficiency towards real-time is an intriguing problem (Yin et al., 2024).

There are also several issues related to the training data. Since we did not include a prompt rewriting procedure in the data curation, the experimental results are focused on relatively short prompts. Also, due to the data filtering procedure, our model did not learn scene transitions during training. This may be overcome by introducing an additional model as the scene director (Lin et al., 2024).

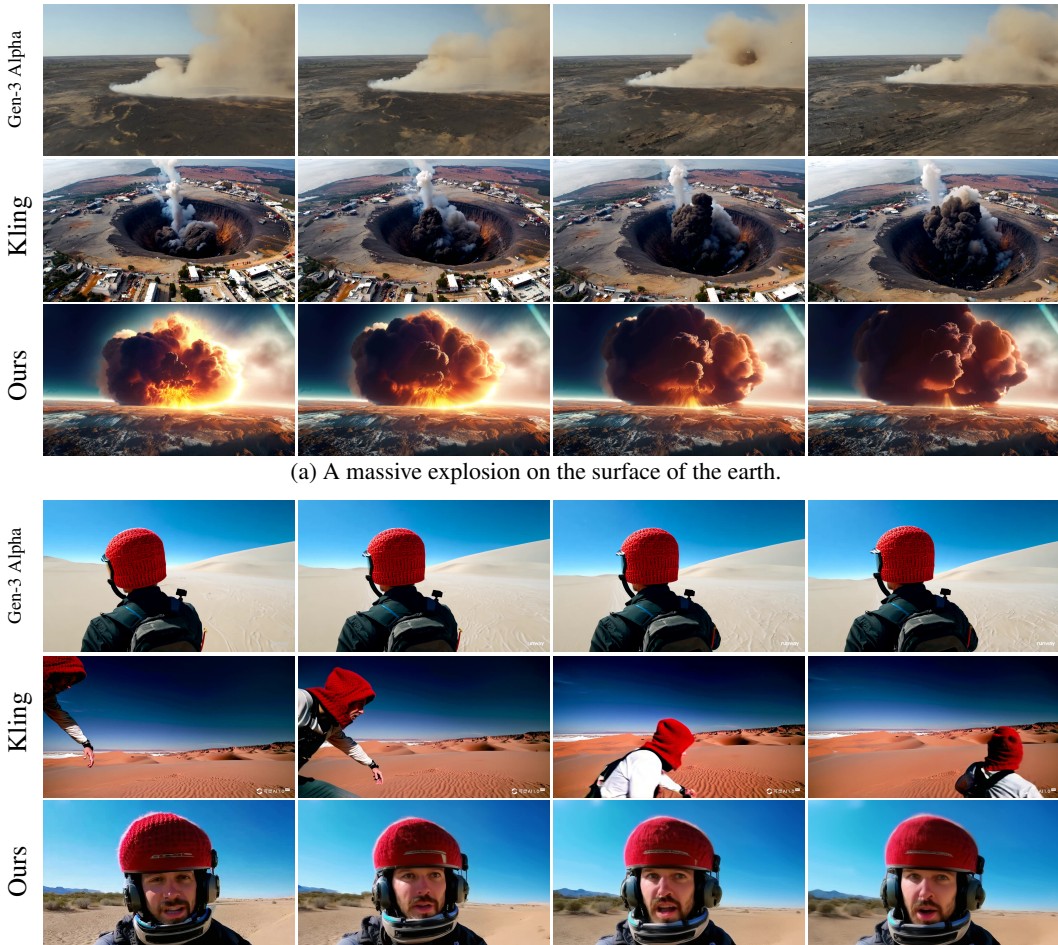

(a) A massive explosion on the surface of the earth.

(b) A movie trailer featuring the adventures of the 30 year old space man wearing a red wool knitted motorcycle helmet, blue sky, salt desert, cinematic style, shot on 35mm film, vivid colors.

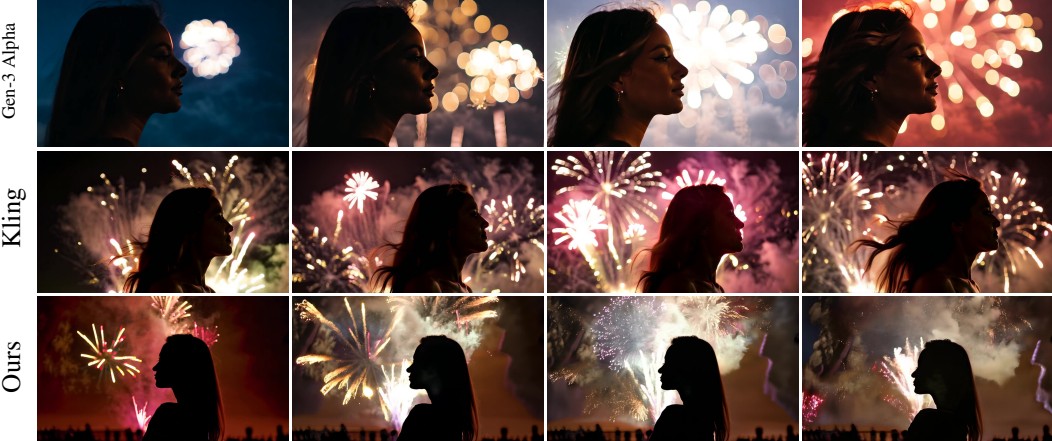

(c) A side profile shot of a woman with fireworks exploding in the distance beyond her.

Figure 14: Visualization of generated videos in comparison with the state-of-the-art closed-source models, including Gen-3 Alpha (Runway, 2024) and Kling (Kuaishou, 2024). Our model delivers cinematic visual quality comparable to these models while adhering to the textual prompt.

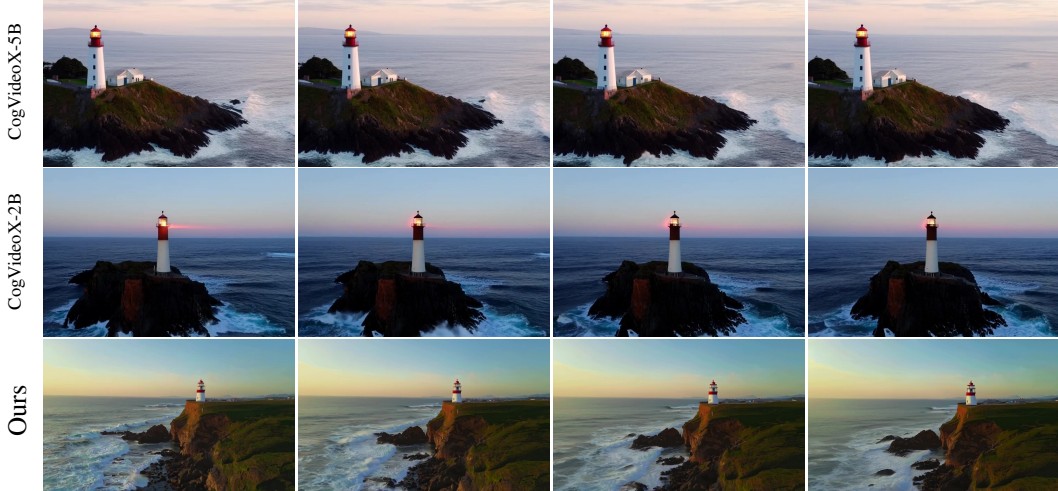

(a) An aerial shot of a lighthouse standing tall on a rocky cliff, its beacon cutting through the early dawn, waves crash against the rocks below.

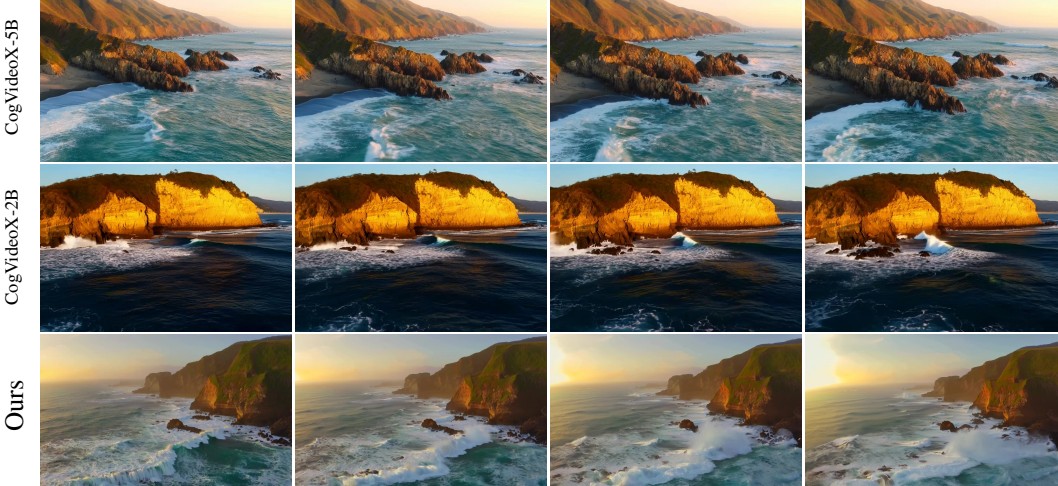

(b) Drone view of waves crashing against the rugged cliffs along Big Sur's garay point beach. The crashing blue waters create white-tipped waves, while the golden light of the setting sun illuminates the rocky shore.

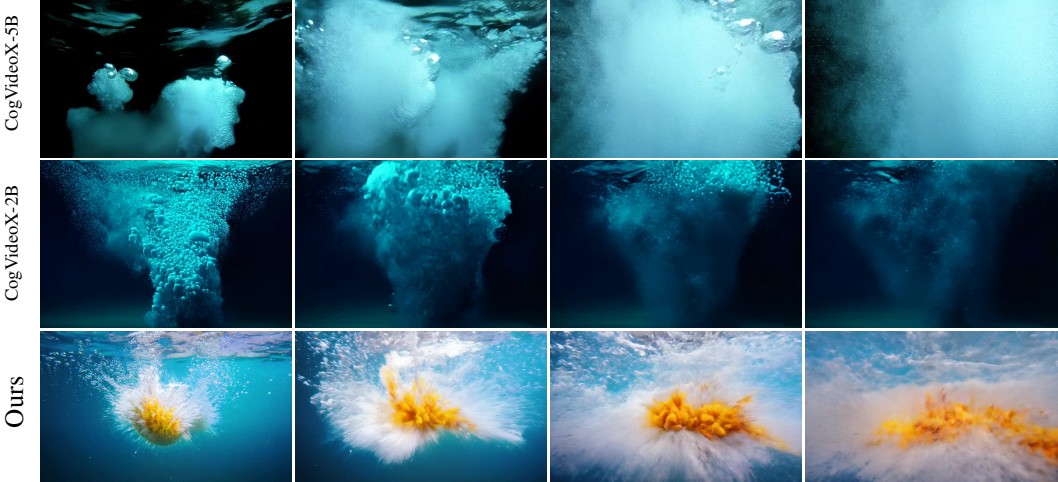

(c) A series of underwater explosions, creating bubbles and splashing water.

Figure 15: Visualization of generated videos in comparison with CogVideoX (Yang et al., 2024). Our model outperforms CogVideoX-2B of the same model size and is comparable to the 5B version.

