# OpenReview forum: "Pyramidal Flow Matching for Efficient Video Generative Modeling"
_ICLR.cc/2025/Conference — ICLR 2025 Poster_

### Official Review · Reviewer_8tU7 · 2024-10-28

**Soundness:** 4
**Presentation:** 2
**Contribution:** 4
**Rating:** 8
**Confidence:** 4

**Summary:**

The authors propose a novel video generation method to address the problems of the current methods. The SOTA (state-of-the-art) methods mostly suffer from heavy computational burdens due to cascaded architectures or separate optimization for different resolutions of video generation training. The authors address this issue with a novel pyramid flow matching framework, both in the spatial and temporal domains. In the spatial domain, the method builds a pyramid of different resolution frames, where the early stages operate on compressed, low-resolution representations (latent representations) of the frames. As the stages go higher up, the resolution increases, and the full resolution is only used in the final stage for optimization. Temporal compression is achieved by using only a lower resolution history of previous frames (to understand motion and scene), further reducing computation for long videos. This pyramidal approach reduces redundant computation by focusing resources only on necessary parts, making training more efficient. The results are comprehensive and satisfactory. Overall, a good paper to be accepted.

**Strengths:**

The computational complexity of SOTA video generation methods is a real problem, and the pyramidal flow matching solution provided by the authors is interesting and effective.

The pyramidal flow matching framework is built on a single diffusion transformer model and can be optimized in a unified, end-to-end fashion. This saves time and enables better knowledge sharing. Better knowledge sharing can help achieve greater consistency in the generated videos, which is evident from the generated results.

Evaluated on benchmarks like VBench and EvalCrafter, the model demonstrated high performance, especially compared to methods trained with open-source data.

The model also achieved competitive performance during the user study, and I have personally checked their provided anonymous website for the videos, which looked good.

**Weaknesses:**

The mathematical notations are not clearly defined. For example, the authors start with s_k, e_k​, x_{s_k}​​, or x_{e_k}​​, etc., without properly defining them first. This hinders the flow of the paper. Please resolve all such issues.

Overall, the writing style of the paper is a bit convoluted (especially in the methods section); it should be revised for smoother understanding.

There is no ablation study on the number of pyramid stages, which is a crucial factor in their design choice.

There is no comparison between the number of parameters and FLOPs used by other open-source methods and the proposed method. This can reflect how effective the model is compared to other methods.

**Questions:**

What are the training data used by the other compared methods (especially the ones trained with open-source data)?

How much overlap is there between the training data used in the proposed method and the other methods?

Why did the authors keep the number of pyramid stages set to 3 in all the experiments? There should be an ablation study on the number of pyramid stages.

What are the number of parameters and/or FLOPs (operations) used by other (open-source) methods for video generation?

---

> ### Author Response · Authors · 2024-11-22
> **Response to Reviewer 8tU7**
>
> We sincerely appreciate the reviewer for recognizing our work and providing valuable feedback. Below are our responses to the raised concerns.
>
> [W1] Mathematical notations.
>
> Sorry for the unclear mathematical notation. We have revised the draft and included the detailed explanations for the used notations in Table 3 of the appendix.
>
> [W2] Writing style.
>
> * Thanks for your valuable suggestions. Due to the limited time in the rebuttal phase, we will try to make the presentation of the methods section clearer in the future. We would appreciate it if you have any other detailed revision advice.
>
> [W3/Q3] Ablation for the number of pyramid stages.
>
> * Following your suggestion, we perform an additional ablation study on image generation (evaluated on MSCOCO) to investigate the influence of pyramid stage numbers by training the model from scratch. As shown in the following Table, using 4 stages can further improve the model's convergence speed. More pyramid stages are not being experimented since the resolution could not be evenly divided.
>
>   | FID at step | 3 Pyramid stages | 4 Pyramid stages |
>   | ----------- | ---------------  | ---------------- |
>   | 10k         | 53.57            | 52.83            |
>   | 20k         | 46.82            | 43.94            |
>   | 30k         | 42.01            | 43.70            |
>   | 40k         | 41.46            | 40.30            |
>   | 50k         | 39.71            | 38.44            |
>
> * In the paper, we set it to 3 to fully utilize low-resolution video data. For example, the videos from WebVid-10M cannot be downsampled 4 times by $2^4$ (since they are already spatially compressed by 8 using the VAE and require a patch size of 2).
>
> [W4/Q4] Comparison of model size and FLOPs.
>
> * The number of parameters and FLOPs are shown in the following table. While our method has a comparable model size to the open-source baselines ($\approx$ 2B), it surpasses them on VBench or EvalCrafter (Tables 1 and 2 in Section 4.3). In terms of FLOPS, we compare with CogVideoX by evaluating the computational FLOPs of each sampling step during generation. It shows that our autoregressive model yields lower FLOPs than the diffusion-based CogVideoX, thanks to its temporal compression designs.
>
>   | Model               | #Parameters | FLOPs (G) | Output video | Speed (sec) |
>   | ------------------- | ----------- | ----- | ---------------- |----------------|
>   | ModelScope          | 1.7B        | -     | -                |-                |
>   | LaVie               | 3B          | -     | -                |-                |
>   | Open-Sora Plan v1.3 | 2.7B        | -     | -                |-                |
>   | Open-Sora 1.2       | 1.1B        | -     | -                |-                |
>   | CogVideoX-2B        | 2B          | 47227 | 720 x 480 x 49   |90                |
>   | CogVideoX-5B        | 5B          | 169192| 720 x 480 x 49   |180                |
>   | CogVideoX 1.5-5B | 5B  |   -    | 1360 x 768 x 81  | 1000 |
>   | Ours (384p 5s)      | 2B          | 30154 | 640 x 384 x 121  |62                |
>   | Ours (768p 5s)      | 2B          | 112386| 1280 x 768 x 121 |336                |
>
> * Note that the main contribution of this paper is to improve training efficiency, as highlighted in Section 4.2. Any other gains in inference efficiency are essentially a byproduct of the careful compression designs.
>
> [Q1/Q2] Comparison of training data
>
> * We summarize the training data used by open-source baselines in the table below. As shown, the baselines are often trained on larger video datasets than ours. Meanwhile, our model outperforms them on Vbench or EvalCrafter (Tables 1 and 2 in Section 4.3). This confirms the data efficiency of our approach.
>
>   | Model                             | #Videos | Video source                 | Overlap with ours |
>   | --------------------------------- | ------- | ---------------------------- | ----------------- |
>   | ModelScope, Show-1, VideoCrafter2 | 10M     | WebVid-10M                   | 10M               |
>   | LaVie                             | 25M     | Vimeo25M                     | $\approx$ 0       |
>   | Open-Sora Plan v1.3               | 19M     | Panda-70M                    | $\approx$ 1M      |
>   | Open-Sora 1.2                     | 30M     | WebVid-10M, Panda-70M, etc.  | $\approx$ 11M     |
>   | CogVideoX                         | 35M     | private                      | unknown           |
>   | Ours                              | 12M     | WebVid-10M, OpenVid-1M, etc. | -                 |

---

> > ### Comment · Reviewer_8tU7 · 2024-11-23
> > **Reply to the authors**
> >
> > After looking at the response on the pyramid ablation study and the flops analysis, I do not have any further concerns about the paper. It would be great have to have clearer method section in the future iterations as promised by the authors. I think it its is a good paper and I will support it.

---

> > > ### Author Response · Authors · 2024-11-24
> > >
> > > Thank you for your kind recognition of our work. We will try our best to polish the expression of method section.

---

### Official Review · Reviewer_BWBa · 2024-11-04

**Soundness:** 2
**Presentation:** 3
**Contribution:** 2
**Rating:** 5
**Confidence:** 4

**Summary:**

This paper introduces pyramidal flow matching, a new video generative model that combines spatial and temporal pyramid representations, which can enhance the training efficiency and maintains high video generation quality.

**Strengths:**

This paper proposes a novel pyramidal flow matching algorithm based on the premise that the initial timesteps in diffusion models are quite noisy and uninformative. This approach builds upon the recent prevalent flow matching framework to address the shortcomings of existing methods, specifically the requirement of employing distinct models at different resolutions, which sacrifices flexibility and scalability.

**Weaknesses:**

1. The authors should clarify the specific implications of how the separate optimization of each sub-stage hinders knowledge sharing and sacrifices flexibility. This is crucial as it relates to the foundational aspects of the problem design presented in this paper.
2. The proposed method involves many parameters that require manual design. The authors should provide more explanations when designing these parameters, such as time windows and γ.
3. Figures 2 and 3 in the paper are overly simplistic, and the authors should provide a detailed explanation of their relationship to the equations.

**Questions:**

1. The authors should clarify whether their proposed method can be applied to text-to-image models, as the method appears to have limited relevance to video generation.
2. The authors should explain why they opted for flow matching over DDIM, as well as outline the advantages of flow matching.

---

> ### Author Response · Authors · 2024-11-22
> **Response to Reviewer BWBa**
>
> We sincerely appreciate the time and effort you dedicated to reviewing our paper and providing valuable feedback. Below are our responses to the raised concerns.
>
> [W1] Implication of separate optimization.
>
> * There are two drawbacks of the previous cascaded diffusion models: (1) Each module specialize only in generation or super-resolution, neglecting the similarity between the two tasks. This leads to worse results than end-to-end training as a joint model, similar to single-task learning vs. multi-task learning, where the latter converges to global optima. (2) They require modification of the ML infrastructure, which limits their flexibility to scale. For example, their generation and super-resolution modules are typically of different sizes, making model sharding and other useful parallelism techniques difficult to apply.
>
> [W2] Details about the number of windows and $\gamma$.
>
> * Our proposed method does not require much manual design, including the parameter $\gamma$ and time windows. As explained in lines 250-252 of page 5, our method derives $\gamma=-1/3$ (to guarantee the positive semi-definite property of covariance matrix) as the theoretically optimal setting (for efficient renoising) and adopts it in all experiments.
>
> * For the number of time windows, we set it to 3 because this is the upper bound supported by our data. For example, the videos from WebVid-10M cannot be downsampled 4 times by $2^4$ (given they are already spatially compressed by 8 using the VAE and require a patch size of 2). However, we expect the use of more time windows to be beneficial to further improve the training convergence. We conduct an ablation study on image generation (MSCOCO) to investigate the influence of pyramid stage numbers by training the model from scratch. As shown in the following Table, using 4 stages can further improve the convergence speed. We did not experiment with more stages since the resolution could not be evenly divided. As for the time window range setting, we simply divide them uniformly and do not use some complicated manual designs.
>
>   | FID($\downarrow$) at step | 3 Pyramid stages | 4 Pyramid stages |
>   | ----------- | ---------------  | ---------------- |
>   | 10k         | 53.57            | 52.83            |
>   | 20k         | 46.82            | 43.94            |
>   | 30k         | 42.01            | 43.70            |
>   | 40k         | 41.46            | 40.30            |
>   | 50k         | 39.71            | 38.44            |
>
> [W3] Connection of figures to the equations.
>
> * We apologize for any confusion about Figures 2 and 3, and explain their connections to the equations below: Figure 2 illustrates the spatial pyramid flow, including its training and inference. Specifically, Figure 2a shows its training, where the flow trajectory is curated by the start and end points in Equations (9) and (10). And Figure 2b provides its inference details, in particular the renoising step in Equation (15) across different stages.
> * Figure 3 illustrates the pyramidal temporal condition. Specifically, Figure 3a shows its compressed history condition as in Equation (17), while Figure 3b shows its position encoding details in lines 315-317 of page 6. We will carefully revise the figures and their captions to reflect these connections.
>
> [Q1] Application to text-to-image generation.
>
> * The proposed method can also be applied to text-to-image generation. The autoregressive video generation model natively generates a high-quality image as the first frame, see Figure 5 for examples. Therefore, the pyramid-flow have the text-to-image generation capability. We have recently trained a 1024px text-to-image generation model from scratch using Pyramid Flow. Even with only a few million training images, it already shows excellent visual quality, see Figure 12(a) for the generated images.
>
> [Q2] Advantages of flow matching.
>
> * We adopt flow matching for its flexibility in interpolating between arbitrary source and target distributions. In contrast, DDIM and other ODE-based diffusion models typically interpolate between the standard Gaussian and data distributions, which prohibits the flexible design of pyramidal flows. In addition, this work has greatly benefited from the simple parameterization and scheduler designs of flow matching, which are crucial for scalable training.

---

### Official Review · Reviewer_csnD · 2024-11-04

**Soundness:** 2
**Presentation:** 3
**Contribution:** 3
**Rating:** 6
**Confidence:** 5

**Summary:**

The paper proposed a pyramid-based flow matching for efficient training of video diffusion models. Unlike traditional flow matching which operates on a single resolution, the proposed spatial pyramidal flow matching operates on a pyramid of resolutions. The denoising process starts from a small resolution, and the resolution will increase by 2 after several sampling steps. To address the discontinuity of the jumping point, the paper proposed a novel re-noising formula. For video generation, the paper proposed an autoregressive approach, where the generation of each frame is conditioned on the low-resolution of previous frames.

**Strengths:**

- The ideas of both spatial pyramids and temporal pyramids are novel and interesting.
- The training efficiency is largely improved due to the novel pyramid design.

**Weaknesses:**

- The analysis of inference efficiency is lacking. How does the proposed method compare to previous full-attention methods for different numbers of frames?
- Compared to full-attention methods, the proposed autoregressive method may encounter the issue of error drifting when the number of frames increases. At how many frames, the proposed method will fail?
- It will be good to include some video results from previous methods on the project page.
- Figure 7 and Figure 8 both show partial results before training convergence. It will be more convincing to show the training graph with more training iterations or converged training behavior.
- The text-video alignment is worse compared to previous methods.

**Questions:**

- During training (Equation-16), noisy condition is used. During testing (Equation-17), clean generated frames are used. Will there be a training-testing distribution mismatch?
- How is text condition added to the model?

---

> ### Author Response · Authors · 2024-11-22
> **Response to Reviewer csnD - Part1**
>
> We sincerely appreciate the time and effort you dedicated to reviewing our paper and providing constructive feedback. Below we clarify the raised concerns one by one.
>
> [W1] Inference efficiency.
>
> * Before presenting the statistics, we note that the main contribution of this work is training efficiency (see Section 4.2) rather than inference efficiency. To evaluate the latter, we compare CogVideoX on a single NVIDIA A100 GPU in terms of total FLOPs and time to generate for different video sizes. For the computational FLOPs, we report the value of one sampling step. It is shown that our autoregressive model yields lower FLOPs and inference time than the diffusion-based CogVideoX, thanks to its pyramidal compression designs.
>
>   | Model            | Output video     | FLOPs(G) | Speed (sec) |
>   | ---------------- | ---------------- | -----   | -----  |
>   | CogVideoX-2B     | 720 x 480 x 49   | 47227  | 90    |
>   | CogVideoX-5B     | 720 x 480 x 49   | 169192 | 180   |
>   | Ours (384p 5s)   | 640 x 384 x 121  | 30154  | 62    |
>   | CogVideoX 1.5-5B | 1360 x 768 x 81  |   -    | 1000  |
>   | Ours (768p 5s)   | 1280 x 768 x 121 | 112386 | 336   |
>
> [W2] Error drifting in autoregressive generation.
>
> * We'd like to share two interesting observations: (1) The proposed method fails after generating 241 video frames (or 31 latent frames), which is exactly the training context length. This is more similar to LLMs where the model performs well within the training context length and fails beyond. (2) Error drifting is indeed a key problem in autoregressive video generation, but it is not specific to causal attention or full attention. For example, Figure 11 shows that autoregressive models perform worse with full attention. Thus, the essence of error drifting requires further investigation as in [1, 2].
>
> [W3] Videos from compared methods.
>
> * Thank you for your valuable suggestion. Indeed, a comparison to the baseline videos on the project page (as in Appendix C.3) improves clarity. We will update the project page in future revisions; it is unclear if this is allowed during rebuttal.
>
> [W4] Baseline with more training iterations.
>
> * To clarify, since our main focus is on training efficiency, most experiments utilize a fixed training budget. We expect that given enough training iterations, all reasonable baselines will converge to similarly good performance, as suggested by the Platonic Representation Hypothesis [3]. However, since training computation is still the performance bottleneck in most scenarios, it is important to investigate training efficiency, as in our work.
>
> [W5] Inferior text-video alignment.
>
> * The reason behind inferior text-video alignment is detailed in Section C.1, mainly due to the data issue:
>
>   > This is largely due to our video captioning procedure based on video LLMs which tends to produce coarse-grained captions, thus dampening these abilities.
>
>   To illustrate this, below are 5 captions sampled from the recaptioned WebVid-10M dataset, which are significantly coarser than the baselines such as CogVideoX and Open-Sora, resulting in inferior text-video alignment. Therefore, we believe that well-captioned video datasets are critical for the development of better video generative models.
>
>   > a bunch of green grapes on a black background
>   >
>   > a dust storm in a city, with buildings barely visible through the sandy air
>   >
>   > an arieal view of a dock with a large ship and a smaller boat
>   >
>   > a man playing a guitar on stage at a concert
>   >
>   > a truck driving on a road in a desert environment
>
> * We have recently trained the pyramid flow from scratch on the same data using the FLUX structure. During training, we filter out the low quality image-text pairs in the LAION dataset. The performance of the new pyramid-flow-miniflux model on VBench is reported in the following table. We find that improving the quality of the captions can significantly improve the text-video alignment even with fewer training iterations.
>
>   | Model            | Total Score   | Quality Score | Semantic Score |
>   | ---------------- | ------------ | -----   | -----  |
>   | Open-Sora Plan v1.1     | 78.00   | 80.91  |  66.38    |
>   | Open-Sora 1.2     | 79.76   |  81.35 |  73.39   |
>   | VideoCrafter2   | 80.44  | 82.20  | 73.42    |
>   | T2V-Turbo | 81.01  |   82.57    | 74.76  |
>   | CogVideoX-2B | 80.91  |   82.18 | **75.83**  |
>   | Ours (SD3)   | 81.72 | **84.74** | 69.62   |
>   | Ours (Miniflux)   | **81.77** | 83.82 | 73.56   |

---

> ### Author Response · Authors · 2024-11-22
> **Response to Reviewer csnD - Part2**
>
> [Q1] Training-test mismatch in autoregressive generation.
>
> * The training-test mismatch is small because the training noise is randomly sampled from [0, 1/3], which covers the test scenario where no noise is added. A similar practice is adopted in [1], where the randomly sampled training noise covers certain noise patterns for testing. We note that one can explicitly remove the training-test discrepancy by adding a conditional flag indicating the noise level [2], but we did not observe a need in our experiment.
> * Nevertheless, how to add noise to the history frames remains a key design issue in autoregressive video generation. We find that using low noise leads to temporal degradation, while high noise overemphasizes the model's ability to generate, causing the model to not strictly follow the history frames. We will continue to work on this issue in autoregressive video generation.
>
> [Q2] Details of text conditioning.
>
> * In terms of model structure, we add text conditions according to MM-DiT [4], namely by joint attention of text and visual features as well as AdaLN. In terms of training, we implement classifier-free guidance that drops the text condition with a probability of 10%, which is known to be essential for the generation quality of diffusion models.
>
> ---
>
> Reference:
>
> [1] Chen, et al. Diffusion forcing: Next-token prediction meets full-sequence diffusion. NeurIPS 2024.
>
> [2] Valevski, et al. Diffusion models are real-time game engines. arXiv preprint arXiv:2408.14837.
>
> [3] Huh, et al. The Platonic representation hypothesis. ICML 2024.
>
> [4] Esser, et al. Scaling rectified flow Transformers for high-resolution image synthesis. ICML 2024.

---

> > ### Comment · Reviewer_csnD · 2024-11-27
> >
> > My concerns have been addressed. I'll keep the positive rating.

---

### Official Review · Reviewer_fYS3 · 2024-11-04

**Soundness:** 4
**Presentation:** 4
**Contribution:** 4
**Rating:** 8
**Confidence:** 5

**Summary:**

This work presents an effective flow matching approach for text-to-video generation via both spatial and temporal pyramidal designs. With such an architecture, the training efficiency has been significantly reduced. It has a good conversation that when noise is strong, flow matching is less critical and can be performed at a low resolution. It has a unified flow matching objective instead of having different models for generation and super-resolution. The paper has validated this effectiveness by controlled experiments compared to a baseline with full-resolution with the same computational costs. The source code of this paper is expected to be open-source, which is very helpful to video generation research and industrial communities.

**Strengths:**

The strength of this paper is multi-fold.
+ It builds a flow matching model with multiple resolutions for text-to-video generation. The pyramidal flow matching allows the model to train with less computational costs and memory footprints.
+ The whole model has a unified objective instead of optimizing separate modules for video generation and super-resolution, using a single Diffusion Transformer.
+ The experimental results are competitive, with evaluation on two public benchmarks of VBench and EvalCrafter. The visual quality is comparable to other commercial text-to-video models.
+ The technical explanation for inference with renoising to solve jump points is clear with supplemental materials.
+ The model can be extended to image-to-video generation.
+ The ablation studies are conducted to illustrate the contribution of different component in the model.

**Weaknesses:**

While this work has an interesting novel design for flow matching for video generation and competitive visual results, there are some unclear points and weaknesses as follows.
- Questions about [s_k, e_k]. The authors divide [0,1] into K time windows [s_k,e_k]. Why don't the authors set e_{k+1}=s_k? Instead, the authors use e_{k+1}=2s_k/(1+s_k) and we can that e_{k+1}>s_k. This means there are overlapping time windows. Given a time step t, t may fall on more than one time step, and how do authors handle such a scenario? Also, is s_K equal to zero? Could you show what the exact values of [s_k,e_k] are?
- The Figure 1(b) and Figure 3 are not consistent. x^i_t takes x^{i-1}^t with the same spatial resolution as part of the history in Figure 3, but Figure 1(b) shows x^i_t takes x^{i-1}_t' with lower resolution as a temporal condition. This part confuses me.
- The model is autoregressive in the temporal domain, and does it mean that the model can generate videos with arbitrary lengths? Why the inference is limited to videos up to 10 seconds?
- When performing the ablation study for the temporal pyramid, the baseline "full-seq" has only qualitative results. Can the authors provide quantitative results or something similar to the plot in Figure 7?

**Questions:**

In the rebuttal, I hope the authors can address my questions and concerns in the weakness section.

---

> ### Author Response · Authors · 2024-11-22
> **Response to Reviewer fYS3**
>
> We sincerely appreciate the reviewer for recognizing our work and providing valuable feedback. Below are our responses to the raised concerns in your review.
>
> [W1] Details of time windows.
>
> * The renoising step ($e_{k+1}\neq s_k$) is derived assuming Equations (7) and (8), see Appendix A for derivation. On the other hand, as you suggested, a flow model with $e_{k+1}= s_k$ can be defined instead by the following endpoints:
>   * End: $\hat x_{e_k}=e_k\mathit{Down}( x_{1},{2^k})+(1-e_k)n$,
>   * Start: $\hat x_{s_k}=\mathit{Up}(s_k\mathit{Down}( x_{1},{2^{k+1}})+(1-s_k)n)$
>
>   In our early experiments, this flow model shows inferior visual quality. We suspect this is because it cannot perform super-resolution and denoising at the same time, since it implies super-resolution first and then denoising. Overall, we recommend using Equations (7) and (8), which results in $e_{k+1}\neq s_k$.
>
> * To avoid ambiguity in overlapping timesteps, we do not compute the timestep embedding based on the original timestep $t$, but on a globally normalized timestep $\frac{t-s_i+\sum_{k>i}(e_k-s_k)}{\sum_k(e_k-s_k)}$, where $i$ is the current pyramid stage.
>
> * The starting point $s_K$ is indeed zero, corresponding to pure noise. The time windows are simply divided with uniformly spaced endpoints, namely $e_k=1-\frac{k}{K}$. According to Equation (26), these time windows are  $[s_k,e_k]=[\frac{K-k-1}{K+k+1},1-\frac{k}{K}]$.
>
> [W2] Ambiguity in temporal condition.
>
> * We apologize for any confusion between Figure 1b and Figure 3. They both illustrate the temporal condition $\{x^0,\ldots,x^{i-1}\}$ that gradually changes in resolution, but Figure 3 additionally shows the current prediction $x^i$. We will revise the figure captions to make this difference clearer.
> * In terms of resolution, your observation based on Figure 3 is correct, namely that $x^{i-1}$ has the same spatial resolution as $x^i$. This is because a latent of the same resolution is necessary to provide visual detail to maintain temporal consistency.
>
> [W3] Autoregressive generation length.
>
> * Our current model cannot generate videos of arbitrary length because it does not utilize sliding windows. Sliding window is a common technique to autoregressively generate longer videos at test time [1, 2], but in this work we focus on training models natively on long videos, rather than adding their support post hoc. Nevertheless, it is possible to combine our 10-second model with sliding windows to generate longer videos.
>
> [W4] Quantitative results for full-sequence diffusion.
>
> * Thanks for your valuable suggestions! We have quantitatively evaluated the FVD metric of the full-sequence diffusion baseline and our pyramid-flow on the MSR-VTT benchmark. The FVD plot along training iterations is illustrated in Figure 12(b) in the appendix. For convenience, the detailed results are also presented in the following Table. As observed, the convergence rate of pyramid-flow is significantly improved compared to the standard full-sequence diffusion.
>
>   | FVD($\downarrow$) at step | full-seq diffusion | Pyramid-Flow (ours) |
>   | ----------- | --------------- | ---------------- |
>   | 10k         |     513.42       |    355.16       |
>   | 20k         |     450.46      |     315.18       |
>   | 30k         |    403.29      |     277.57       |
>   | 40k         |    370.25        |   209.49         |
>   | 50k         |   310.47    |    165.52         |
>
> ---
>
> Reference:
>
> [1] Chen, et al. Diffusion forcing: Next-token prediction meets full-sequence diffusion. arXiv preprint arXiv:2407.01392.
>
> [2] Valevski, et al. Diffusion models are real-time game engines. arXiv preprint arXiv:2408.14837.

---

> > ### Comment · Reviewer_fYS3 · 2024-11-26
> >
> > Thanks very much for your detailed explanation and the revision to the paper. I think my concerns are addressed and will keep the same positive rating.

---

> > > ### Author Response · Authors · 2024-11-27
> > >
> > > Thank you for your valuable suggestions, which greatly improve the quality of our work.

---

### Official Review · Reviewer_6mLb · 2024-11-05

**Soundness:** 2
**Presentation:** 2
**Contribution:** 3
**Rating:** 8
**Confidence:** 4

**Summary:**

This paper introduces a novel pyramidal flow matching scheme for video generation, which significantly improves training efficiency while preserving generation quality. The authors also propose a unified flow-matching objective that enables joint training of the pyramid stages within a single DiT model, eliminating the need for separate optimization across multiple models seen in prior approaches. Comprehensive experimental analyses are conducted on the VBench and EvalCrafter benchmarks, with proofs and additional qualitative results included in the supplementary materials.

**Strengths:**

+ The proposed pyramidal flow matching scheme is novel in video generation modeling and greatly enhances training efficiency.
+ The unified training objective is intuitive and effective.
+ The quantitative and qualitative analyses in the paper are comprehensive.
+ The quality of the generated videos is excellent.

**Weaknesses:**

* The writing in the paper could benefit from further improvement for clarity and readability.
    - The repeated use of the term "full-resolution" up to the experiment section suggests that generation is being done in pixel space rather than latent space. It would be helpful to clarify this in the paper, as it may be misleading.

    - The paper contains several grammatical errors and repeatedly uses unnecessary terms, such as "sophisticated," which affect readability. I encourage the authors to revise the manuscript to improve clarity and flow.

* Some of the derivations and assumptions in the paper are ambiguous and potentially flawed

    - In Equations 7 and 8, the notation should reflect the conditional distributions $\hat{x}{e_k}|x_1$ and $\hat{x}{s_k}|x_1$. While both endpoints are Gaussian-distributed conditionally, the endpoint distributions $\hat{x}{e_k} = \int p(\hat{x}{e_k}|x_1)p(x_1)dx_1$ are not Gaussian.

    - In line 213, while the objective $\hat{x}{e_k} - \hat{x}{s_k}$ is correct, since we consider flow matching with $K$ windows, the vector field should instead be conditioned on the endpoints, $u_t(x_t|\hat{x}{e_k})$. However, this is challenging because the distribution $p(x_t|\hat{x}{e_k})$ may not be Gaussian. This objective could be derived more straightforwardly from a rectified flow perspective [1], where velocities are matched with $\dot{X}_t$ (see section 2.3 in [1]).

    - For training, are $x_1$ and $n$ at the start and end the same? Since flow matching accommodates data points from arbitrary couplings, this should not impact training validity but would benefit from additional explanation for clarity.

    - Regarding the renoising procedure in Section 3.2.2, Equation 12 should ideally denote $Up(\hat{x}{e_{k+1}}|x_1)$ rather than $Up(\hat{x}{e_{k+1}})$, as the latter may be non-Gaussian. Consequently, the subsequent proof may be invalid if it relies on $Up(\hat{x}{e_{k+1}})$ being Gaussian.

         [1] Flow Straight and Fast: Learning to Generate and Transfer Data with Rectified Flow

* Some important details and experiments are missing from the paper.

    - In Sec 3.2.1, the authors claim to enforce the noise to be in the same direction to enhance the straightness of the flow trajectory (Lines 207-209). However, there is no ablation experiment to show the benefit of this design choice.

    - The proof in Section 3.2.2 and Appendix A primarily relies on Equation 8. However, the authors implement the scheme in Equation 10, with no discussion on how the derivations in Section 3.2.2 and Appendix A could be generalized for Equation 10.
    - What is the number of sampling steps in Algorithm 1?

    - (Minor) Have the authors considered using temporally compressed pyramidal flow as well?


* No experimental results are provided for the video autoencoding task
   - In Section 4.1, the authors claim to train a video VAE for spatial and temporal compression of videos; however, there are no experimental results provided to evaluate the performance of the video VAE.
    - Why train a new video VAE instead of utilizing existing open-source video VAEs, such as Open-Sora or Open-Sora-Plan? Wouldn't this approach have been more effective in ensuring a fair experimental comparison?

* The results on VBench and EvalCrafter are quite strong. However, since most of the competing approaches were published prior to the release of these benchmarks, it would be beneficial for the authors to include the FVD metric to compare their approach with the competing baselines. Additionally, could the authors provide a comparison of the generation and training speeds of their approach relative to previous works?

**Questions:**

Please refer to the questions (or issues) mentioned in the Weaknesses section.

---

> ### Author Response · Authors · 2024-11-22
> **Response to Reviewer 6mLb - Part1**
>
> We sincerely appreciate the time and effort you have taken to review our paper and provide valuable feedback. Below are our responses to the concerns raised in your review.
>
> [W1] Clarity of writing.
>
> * Thanks for pointing that out. The "full resolution" refers to the vae latents used in traditional latent diffusion models (LDM), and the generation of our pyramid flow is done in latent space, not pixel space. We apologise for the misleading wording. We have revised the draft to clarify this and to change the grammatical errors or unnecessary terms, following your valuable suggestions.
>
> [W2] Theoretical derivation.
>
> * Thank you for the valuable suggestion. In the line before Equations (7) and (8), we mentioned "conditional probability path", which implies that the following two equations are conditioned on $x_1$. To avoid any ambiguity, we have revised the draft to state this explicitly. Note that this does not affect subsequent derivations, e.g. Equation (13), because it only involves taking the expectation on both sides of the equation.
> * One thing to clarify is that the conditional flow within each window is actually conditioned on the real endpoint $x_1$ instead of the window endpoint $\hat{x}_{e_k}$. By this definition, the conditional distribution should be Gaussian.
> * Compared to rectified flow, flow matching learns random couplings between the data $x_1$ and the noise $n$ (they are not the same distribution), which is less inference-efficient due to curved flow trajectories. However, rectified flow requires simulation using the flow ODE to derive optimal coupled training samples, which is much less training-efficient.
>
> [W3-1] Benefit of coupling noise.
>
> * The rationale for improving straightness by coupling noise is as follows: The straightness of the flow trajectory is usually compromised when there are intersections. Sampling the endpoints independently (as in vanilla flow matching) creates random directions for each trajectory and leads to intersections. Instead, by coupling the sampling of these endpoints, as in Equations (9) and (10), we can create more organised, possibly parallel trajectories with fewer intersections, thus improving straightness.
> * We further illustrate this with a toy experiment in Figure 13 in the appendix, where coupling noise indeed leads to more straight flow trajectories.
>
> [W3-2] Clarification of Equation (10).
>
> * To clarify, Equation (10) is just an instantiation of Equation (8) that is used only in training. On the other hand, Equation (8) defines the entire flow model and is used in both training and inference. Therefore, the inference procedure in Section 3.2.2 and Appendix A is derived based on Equation (8) only.
>
> [W3-3] Number of sampling steps.
>
> * The number of sampling steps for each stage is set to 10. We have found that this setting achieves a good balance between total inference time and generation quality.
>
> [W3-4] Temporally compressed pyramid flow.
>
> * Temporally compressed pyramid flow should also work. It is not adopted in the paper because next-frame prediction is a more natural choice than next-scale prediction for video. The video frames are already well ordered along the time axis, and there is causality that facilitates autoregressive generation.
>
> [W4] Clarification of video VAE.
>
> * We did not evaluate the quantitative performance of the video VAE because it is not our main technical contribution, the whole architecture just follows MAGVIT-v2 [1]. We have compared our causal video VAE with other open source versions on $256 \times 256$ resolution 17-frame videos from the WebVid. From the results presented, we can see that our VAE achieves a comparable PSNR value to that of CogVideoX with a higher compression rate.
>
>   | Model           | Compression    |PSNR($\uparrow$)|
>   | --------------- | -------------- |-------------- |
>   | Open-Sora       | $8 \times 8 \times 4$| 28.5   |
>   | Open-Sora-Plan  | $8 \times 8 \times 4$| 27.6   |
>   | CogVideoX       | $8 \times 8 \times 4$| **29.1**  |
>   | Ours            | $8 \times 8 \times 8$| 28.9  |
>
> * We train a new video VAE primarily because the normalization design of open-source VAEs is not compatible with our pipeline. To natively support both T2V and I2V, the first frame of the video VAE latent should be identical to an image latent. A simple trick to ensure this is to normalize the first frame and subsequent frames separately, which has not been adopted in open source VAEs.

---

> ### Author Response · Authors · 2024-11-22
> **Response to Reviewer 6mLb - Part2**
>
> [W5-1] FVD metric.
>
> * Thank you for your constructive comments. The following table compares our model with previous baselines on the FVD metric on MSR-VTT, which is a commonly used benchmark to evaluate video generation performance. The results show that our model outperforms the competing baselines, demonstrating the effectiveness of pyramid flow.
>
>   | Model           | FVD on MSR-VTT ($\downarrow$) |
>   | --------------- | -------------- |
>   | CogVideo        | 1294           |
>   | VideoComposer   | 580            |
>   | VideoPoet       | 213            |
>   | Video-LaVIT [2] | 188.36         |
>   | Ours            | **142.83**         |
>
> [W5-2] Inference and training speed.
>
> * We first describe the training speed, which is the main contribution of our work. This has been validated by comparison with the open source baseline (Section 4.2) and by ablation studies (Figures 7 and 8 in Section 4.4). Below is a detailed summary of training costs, where our method shows a significant improvement in training efficiency:
>
>   | Model               | Output video     | GPU hours                |
>   | ------------------- | ---------------- | ------------------------ |
>   | Open-Sora Plan v1.2 | 1280 x 720 x 93  | 37.8k H100 + 4.8k Ascend |
>   | Open-Sora 1.2       | 1280 x 720 x 102 | 35k H100                 |
>   | Ours (768p 10s)     | 1280 x 768 x 241 | 20.7k A100               |
>
> * Next, we compare the video generation inference time and FLOPs on a single NVIDIA A100 GPU with diffusion-based CogVideoX. While our model outperforms CogVideoX on VBench (see Table 1), it yields lower FLOPs and inference time for similar video sizes, demonstrating the superior inference efficiency of pyramid flow. Note that the gain in inference efficiency is essentially a by-product of the autoregressive designs to improve training efficiency.
>
>   | Model            | Output video     | FLOPs(G) | Speed(sec) |
>   | ---------------- | ---------------- | -----   | -----  |
>   | CogVideoX-2B     | 720 x 480 x 49   | 47227  | 90    |
>   | CogVideoX-5B     | 720 x 480 x 49   | 169192 | 180   |
>   | Ours (384p 5s)   | 640 x 384 x 121  | 30154  | 62    |
>   | CogVideoX 1.5-5B | 1360 x 768 x 81  |   -    | 1000  |
>   | Ours (768p 5s)   | 1280 x 768 x 121 | 112386 | 336   |
>
>
> ---
>
> Reference:
>
> [1] Yu, et al. Language model beats diffusion - Tokenizer is key to visual generation. ICLR 2024.
>
> [2] Jin, et al. Video-LaVIT: Unified video-language pre-training with decoupled visual-motional tokenization. ICML 2024.

---

> > ### Comment · Reviewer_6mLb · 2024-11-25
> >
> > I thank the authors for their well-written rebuttal. Most of my concerns have been sufficiently addressed, and I am pleased to raise my score in support of accepting this good work.

---

> > > ### Author Response · Authors · 2024-11-25
> > >
> > > Thank you for your constructive suggestions and kind recognition of our work!

---

### Author Response · Authors · 2024-11-22
**Response to All Reviewers**

We sincerely appreciate all the reviewers for their thoughtful and constructive feedback. We have revised the manuscript and added clarifications based on the reviews. The detailed revision we made is summarized as follows:

* Modify the unclear expressions and words mentioned by Reviewer 6mLb and fYS3.

* Add the detailed explanations for the mathematical notations suggested by Reviewer 8tU7.

* Add the ablation results and analysis of coupling noise suggested by Reviewer 6mLb.

* Add the quantitative results comparsion with full-sequence diffusion baseline suggested by Reviewer fYS3.

* Add the text-to-image generation results mentioned by Reviewer BWBa.

These changes have been highlighted in brown font.

---

### Meta-Review · Area_Chair_KZRM · 2024-12-22

**Metareview:**

The paper presents an important step towards efficient training of text-to-video generation. Such models are known to be very hard to train, requiring a substantial computational budget. Here, we are presented with a method to reinterpret denoising trajectory as a series of pyramid stages, leading to an efficient solution. The reviewers give a very positive assessment of the work listing many strengths of the work. The AC agrees. Congrats!

**Additional Comments On Reviewer Discussion:**

There was a very good back-and-forth between the reviewers and the authors. Reviewer 6mLb shared a number of weaknesses with the authors, including some questions about theoretical derivation. The authors were able to successfully address them. Reviewer BWBa gave a negative rating, listing several questions, in particular whether the the method can be used for text-to-image models. AC believes that the authors adequately addressed the concerns, however, the reviewer didn't get back to the discussion. Other reviewers were quite happy with the work after the discussion period.

---

### Decision · Program_Chairs · 2025-01-22

Accept (Poster)